# Sea Ice Extraction via Remote Sensing Imagery: Algorithms, Datasets, Applications and Challenges

**Wenjun Huang, Anzhu Yu \***　**, Qing Xu, Qun Sun, Wenyue Guo, Song Ji, Bowei Wen and Chunping Qiu**

Institute of Geospatial Information, Information Engineering University, Zhengzhou 450001, China; huangwj_geo@126.com (W.H.); xq1982_no.1@163.com (Q.X.); 13503712102@163.com (Q.S.); guowyer@163.com (W.G.); jisong_chxy@163.com (S.J.); gis0829@infu.ac.cn (B.W.); chunping.qiu@aliyun.com (C.Q.)
* **Correspondence:** anzhu_yu@126.com

**Abstract:** Deep learning, which is a dominating technique in artificial intelligence, has completely changed image understanding over the past decade. As a consequence, the sea ice extraction (SIE) problem has reached a new era. We present a comprehensive review of four important aspects of SIE, including algorithms, datasets, applications and future trends. Our review focuses on research published from 2016 to the present, with a specific focus on deep-learning-based approaches in the last five years. We divided all related algorithms into three categories, including the conventional image classification approach, the machine learning-based approach and deep-learning-based methods. We reviewed the accessible ice datasets including SAR-based datasets, the optical-based datasets and others. The applications are presented in four aspects including climate research, navigation, geographic information systems (GIS) production and others. This paper also provides insightful observations and inspiring future research directions.

**Keywords:** sea ice extraction; semantic segmentation; survey; remote sensing; mapping; machine learning; deep learning

## 1. Introduction

Sea ice extraction (SIE) has been a crucial problem in many application aspects, such as the polar navigation [1], terrain analysis [2], polar cartography [3] and polar expedition [4]. With the rapid development of the machine learning technique, computational capability and data acquisition, the SIE problem has reached the deep learning era. Machine learning-based approaches are being increasingly introduced to detect, segment or map the sea ice.

As a branch of machine learning, the deep learning technique attracts more attention to solve the SIE problem in last five years, based on which the mapping or cartography problem could also be solved. Most of the literature converts the SIE problem to another common topic, namely the semantic segmentation problem, which determines the category of each pixel via a post-classification procedure after the category probability is regressed by deep convolutional neural networks. In recent years, there has been a growing body of research focusing on SIE. To gain deeper insights into this field, we conducted a literature search using the keywords "sea ice extraction" via the Web of Science Core Collection. As illustrated in Figure 1, there has been a notable rise in publications since 2016, prompting our emphasis on summarizing and synthesizing the literature published after this period. Additionally, we utilized the Citespace [5] statistical algorithm to visualize the co-citation network of relevant publications from the past five years (Figure 2). The visualization highlights key themes and research areas associated with SIE, with a particular emphasis on remote sensing and Synthetic Aperture Radar (SAR). Currently, SIE primarily relies on remote-sensing techniques such as visible/infrared remote sensing, passive microwave remote sensing and active microwave remote sensing [6]. Visible/infrared remote sensing can provide texture information of sea ice, which is helpful for SIE tasks. However, it has

certain limitations. Firstly, it is restricted in polar regions due to the occurrence of polar day and polar night phenomena. Additionally, the orbital inclination (typically 97–98°) and altitude of conventional remote-sensing satellites affect observations in polar regions, leading to polar data gaps where effective observations are not possible. Consequently, polar orbit satellites are relied upon for conducting observations. On the other hand, passive microwave remote sensing offers global coverage capabilities and, therefore, holds certain advantages. Nevertheless, its drawback lies in relatively low spatial resolution. Typical instruments for passive microwave remote sensing, such as the Advanced Microwave Scanning Radiometer for EOS (AMSR-E), generally provide spatial resolutions at the kilometer level. Such lower resolutions may not fulfill the requirements for detailed SIE and further mapping. In contrast, active microwave remote-sensing techniques, such as SAR, offer higher-resolution capabilities. SAR technology can achieve resolutions at the meter level, making it highly suitable for fine-scale sea ice mapping [7,8]. As a consequence, current research on SIE predominantly focuses on the application of active microwave remote-sensing technologies, notably SAR. Moreover, significant achievements have been made in SIE tasks through the utilization of optical remote sensing [9] and the integration of SAR with optical approaches [10–12]. In addition to the aforementioned remote-sensing satellite observations, some studies have utilized real-time ice monitoring using aerial images captured by cameras onboard icebreakers [13,14] and unmanned aerial vehicles (UAVs) [15,16]. These methods serve as valuable supplementary approaches for SIE tasks.

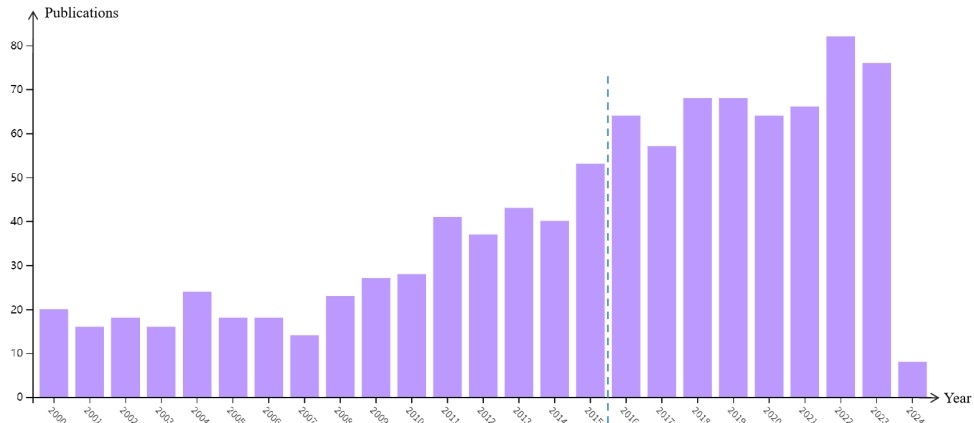

**Figure 1.** Publication trends from 2000 to present in the Web of Science Core Collection. Following the time period marked by the blue dashed line, there is a significant surge in the number of publications, with research during this period constituting the focal point of the review.

Machine learning methods have made significant applications in the field of SIE. Recently, several reviews have provided summaries of sea ice remote sensing. In [17], the focus was on analyzing the advantages and disadvantages of sea ice classification methods based on SAR data. The advancements of Global Navigation Satellite System-Reflectometry (GNSS-R) data in SIE, ice concentration estimation, ice type classification, ice thickness inversion and ice elevation were reviewed. In [8], a comprehensive analysis of sea ice sensing using polarimetric SAR data was conducted. Key geophysical parameters for SIE, including ice type, concentration, thickness and motion, as well as SAR scattering characteristics analysis, were summarized. However, these papers primarily focused on providing overviews of sea ice monitoring methods using SAR technology, lacking comprehensive summaries of specific technical approaches. Moreover, they predominantly concentrated on summarizing sea ice remote-sensing methods and lacked a comprehensive overview of downstream tasks related to SIE, specifically applications. Therefore, this review aims to provide a comprehensive summary of the latest SIE methods developed in the past five years. It aims to systematically categorize and analyze these methods, taking

into account the associated datasets and subsequent mapping applications. Additionally, this review incorporates the latest advancements in technology to assess the challenges and future developments in SIE through the utilization of large-scale models.

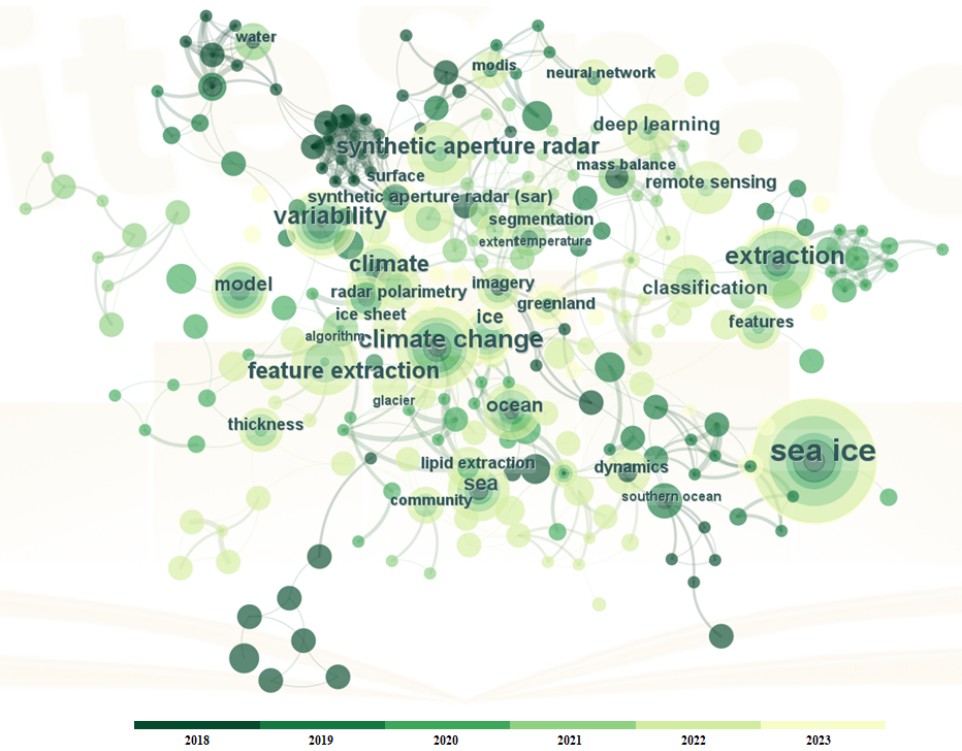

**Figure 2.** The co-citation network for SIE research. The frequency of the keywords was visually represented by the size of the nodes, while the strength of their relationships was indicated by the width of the linking lines. Additionally, the publication year was visually depicted through the color variation of the nodes.

The overall structure of this review is presented in Figure 3. Section 2 of this review will provide detailed insights into recent methods for SIE. Section 3 will summarize the currently available open-source datasets related to ice. Section 4 aims to outline downstream tasks and enumerate the generated geospatial information products resulting from ice extraction. Lastly, Section 5 will highlight areas where future developments are needed.

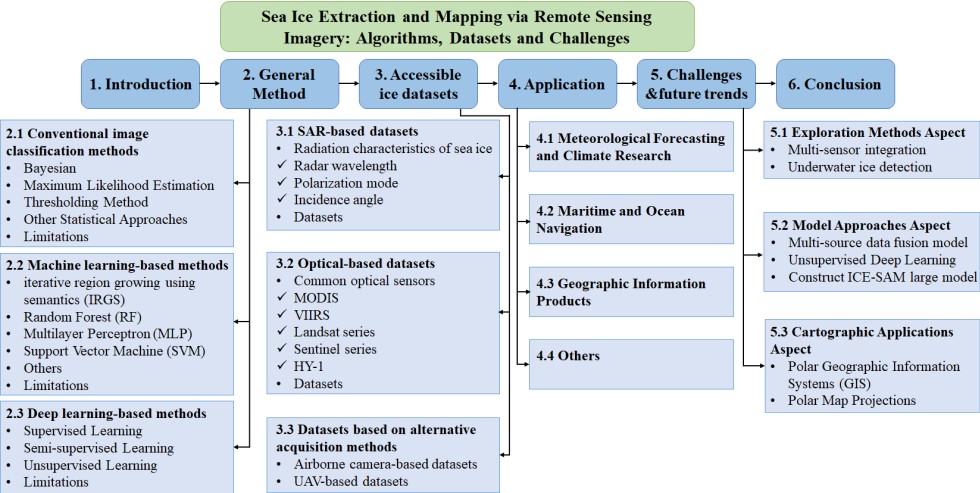

**Figure 3.** Structure of this review.

## 2. Method of Sea Ice Extraction

### 2.1. Conventional Image Classification Methods

In the early stages, research on sea ice concentration (SIC) and SIE primarily relied on statistical algorithms. These algorithms generally combined probabilistic models and classical classification methods with texture or polarization features to generate sea-ice-type maps. There is a rich body of literature on conventional image classification methods, and this section will focus on some recent publications.

#### 2.1.1. Bayesian

A new Bayesian risk function is proposed in [18] to minimize the likelihood ratio for polarimetric SAR-data-supervised classification. A novel spatial criterion is also introduced to incorporate spatial contextual information into the classification method, achieving a sea ice classification accuracy of 99.9%. Bayesian theorem, as described in [19], is utilized to compute the posterior probabilities of each class at each observed location based on the texture features extracted from the gray-level co-occurrence matrix (GLCM) of the image. In [20], the authors label each pixel in the SAR imagery as ice or water using MAp-Guided Ice Classification (MAGIC) [21] and model the labeled pixels as a Bernoulli distribution. The estimated ice concentration is then obtained by incorporating the labeled data into the Bayesian framework along with AMSR-E ice concentration data. The authors of [22] introduce a Gaussian Incidence Angle (GIA) classifier for sea ice classification, which replaces the constant mean vector in the multivariate Gaussian probability density function (PDF) of the Bayesian classifier with a linearly varying mean vector. The simplicity and fast processing time of the GIA classifier enable near real-time ice charting. The authors of [23] utilize this GIA classifier to generate classified winter time series of sea ice in the regions covered during the Multidisciplinary drifting Observatory for the Study of Arctic Climate (MOSAiC) campaign, providing reliable support for navigation.

#### 2.1.2. Maximum Likelihood Estimation

In [24], Maximum Likelihood Estimation is used to compute the probabilities of ice and water in the observed SAR images. An unsupervised mixture Gaussian segmentation algorithm is proposed in [25], which provides reasonable sea ice classification results under similar incidence angle conditions. The authors of [26] apply logistic regression (LR) statistical techniques to demonstrate that the average and variance of texture features, specifically the GLCM, are most suitable for maximum likelihood supervised classification, thus extracting the sea ice density map of the western Antarctic Peninsula region.

#### 2.1.3. Thresholding Method

Zhu et al. [27] utilized the Delay-Doppler Map (DDM) of the Global Navigation Satellite System (GNSS) signals reflected by sea ice and seawater, which exhibit distinct scattering characteristics. The differential DDM, observed as the difference between two adjacent normalized DDMs, provides information about the differences between the two DDMs. By employing a thresholding method, the type of the reflecting surface can be determined, thus extracting the sea ice. Building upon this, Alexander et al. [28] proposed an adaptive probability threshold for automatic detection of ice and open-water areas. The textural and edge features of different sea-ice-types in various turbid regions were discussed, using the Yellow River Delta as an example, laying the foundation for the classification of sea-ice-types. Automatic extraction of sea ice can be achieved by employing the OTSU algorithm to determine the threshold automatically.

#### 2.1.4. Other Statistical Approaches

Additionally, Zhang et al. [29] proposed an automatic classification method for SAR sea ice images combining Retinex and the Gaussian Mixture Model algorithm (R-gmm). Experimental results demonstrated that this algorithm effectively enhances the clarity of SAR imagery compared to the Single-Scale Retinex Algorithm, GMM, and Markov Random

Field (MRF)-based methods, thereby improving segmentation accuracy. In [30], a multi-scale strategy of the curvelet transform was further utilized to extract curve-like features from SAR images, distinguishing the MIZ from open water and consolidated ice areas. Xie et al. [31] employed the polarization ratio (PR) between VV and HH in SAR images calculated based on the roughness characteristics of the sea surface scattering and the X-Bragg backscatter model. This measurement comparison can differentiate between open water and sea ice, achieving an overall accuracy of approximately 96%. Mary et al. [32] utilized the coefficient of variation (COV) from co-pol/cross-pol SAR data to detect thin ice during the Arctic freezing period using a synergistic algorithm.

### 2.1.5. Limitations

Generally, when environmental changes are inevitable, it has been demonstrated that relying solely on conventional image classification methods for real-time object detection is inefficient. In addition to the challenges associated with threshold selection in conventional image classification methods, other classical statistical methods, such as Bayesian and maximum likelihood methods, also exhibit limitations. For instance, Bayesian methods may face challenges related to the assumption of prior probabilities, which might not accurately represent the true underlying distribution of the data. Similarly, maximum likelihood methods can be sensitive to outliers in the data, leading to biased parameter estimates. These limitations highlight the need for more robust and adaptive approaches, particularly in dynamic and unpredictable environments.

While conventional methods have their strengths, these limitations pave the way for exploring alternative approaches to address the aforementioned challenges. By leveraging advanced techniques such as machine learning, probabilistic models, and adaptive algorithms, researchers have sought to overcome the issues associated with threshold-based segmentation. These alternative methods offer promising avenues to enhance segmentation accuracy, handle complex scenes, and mitigate the sensitivity to brightness and noise.

### 2.2. Machine Learning-Based Methods

Machine learning methods primarily leverage the polarimetric characteristics of sea ice images (HH, HV, HH/HV) and selected features such as GLCM texture features. These features are then subjected to rule-based machine learning methods for classification, enabling the differentiation between sea ice and open-water areas. Furthermore, in the literature, there are approaches that further refine the classification of sea ice, distinguishing between multi-year ice (MYI) and first-year ice (FYI), among other categories. Expanding on the various methodological approaches, let us delve into each method and its specific contributions in sea ice classification.

### 2.2.1. Iterative Region Growing Using Semantics (IRGS)

Yu et al. [22] proposed an image segmentation method called IRGS. IRGS [33] models the backscatter characteristics using Gaussian statistics and incorporates a Markov random field (MRF) model to capture spatial relationships. It is an unsupervised classification algorithm that assigns arbitrary class labels to identified regions, with the mapping of class labels left for manual intervention by human operators. Building upon IRGS, several pieces of research have been conducted for sea ice–water classification. Clausi et al. [21] developed a binary ice–water classification system called MAGIC. Subsequently, Leigh et al. [34] used glocal IRGS to capture the spatial contextual information of RADARSAT-2 SAR images and identified homogeneous regions using a hierarchical approach. Pretrained SVM models were then used to assign ice–water labels. The IRGS method, combined with modified energy functions and the contributions of glocal and SVM classification results, balanced the contextual and texture-based information. Ghanbari et al. [35] tested the method with four different SAR data types: dual-polarization (DP) HH and HV channel intensity images, compact polarimetric (CP) RH and RV channel intensity images, all derived CP features, and quad-polarimetric (QP) images. The experimental results demonstrated that

utilizing CP data achieved the best classification results, which were further supported by similar findings in [36,37]. The self-training IRGS (ST-IRGS) was introduced in [38], which integrated hierarchical region merging with conditional random fields (CRF) to iteratively reduce the number of nodes while utilizing edge strength for classification and region merging. The key feature of ST-IRGS is the embedded self-training procedure. Wang et al. [39] extensively tested IRGS on dual-polarization images for lake ice mapping, minimizing the impact of the incidence angle. The experimental results demonstrated that the IRGS algorithm provides reliable ice–water classification with high overall accuracy.

As emerging image classification methods advance, IRGS has been seamlessly integrated with various classification techniques to enhance sea ice classification. Hoekstra et al. [40] integrated IRGS segmentation with supervised labeling using RF. The IRGS segmentation algorithm incorporated spatial context and texture features from the ResNet, utilizing region pooling for ice–water classification [41]. Jiang et al. [42] made a comparison between two benchmark pixel classifiers, SVM and RF, and two models, IRGS-SVM and IRGS-RF. The experimental results indicated that IRGS-RF achieved better performance and demonstrated stronger robustness. In [43], the IRGS algorithm was utilized to oversegment the input HH/HV scene into superpixels. A graph was constructed on the superpixels, and node features were extracted from the HH/HV images. With limited labeled data, a two-layer graph convolution was employed to learn the spatial relationships between nodes. Chen et al. [44] combined the segmentation results from the IRGS algorithm with pixel-based predictions from the Bayesian Convolutional Neural Network (CNN), and by analyzing the uncertainty of SAR images, sea ice and water were distinguished.

These works demonstrate the versatility of IRGS and its integration with different classification methodologies, leading to improved performance and enhanced classification accuracy in sea ice analysis.

### 2.2.2. Random Forest (RF)

Han et al. [45] utilized texture features from backscatter intensity and GLCM as input variables for sea ice mapping and developed a high-spatial-resolution summer sea-ice-mapping model for KOMPSAT-5 EW SAR images using a RF model. Mohammed Dabboor et al. [46] employed the RF classification algorithm to identify effective compact polarimetric (CP) parameters and analyzed the discriminatory role of CP parameters for distinguishing between FYI and MYI. Alexandru Gegiuc et al. [47] applied RF for estimating the ridge density of sea ice in C-band dual-polarization SAR images. Han et al. [48] evaluated four representative sea ice algorithms using binary classification with RF based on PM-measured sea ice concentration (SIC) data. Tan et al. [49] employed a RF feature selection method to determine optimal features for sea ice interpretation and implemented a semi-automated sea ice segmentation workflow. Dmitrii MURASHKIN et al. [50] utilized a RF classifier to investigate the importance of polarimetric and texture features derived from GLCM for the detection of leads. James V. Marcaccio et al. [51] employed image object segmentation and a RF classifier for automated mapping of coastal ice, indicating Laurentian Great Lakes winter fish ecology. Yang et al. [52] developed a RF model to extract lake ice conditions from land satellite imagery. Jeong-Won Park et al. [53] performed noise correction on dual-polarization images, supervised texture-based image classification using the RF classifier, and achieved semi-automated SIE. Meanwhile, in [54], the first approach directly utilizing operational ice charts for training classifiers without any manual work was proposed based on RF.

These studies demonstrate the diverse applications of RF in sea ice analysis, including sea ice mapping, classification of different ice types, feature selection, noise correction and automated ice detection. The RF model has shown its effectiveness in leveraging various image features for accurate and efficient sea ice analysis and has contributed to advancements in sea ice research and monitoring.

### 2.2.3. Multilayer Perceptron (MLP)

Ressel et al. [55] compared the polarimetric backscattering behavior of sea ice in X-band and C-band SAR images. Extracted features from the images were input into a trained Artificial Neural Network (ANN) for SIE. The experiments found that the most useful classification features were matrix-invariant features such as geometric strength, scattering diversity and surface scattering fraction. In [56], further evidence was presented for the high reliability of neural network classifiers based on polarimetric features, demonstrating their suitability for near real-time operations in terms of performance, speed and accuracy. The authors of [57] used neural networks to describe the mapping between image features and ice–water classification, with texture features extracted from co-polarized and cross-polarized backscatter intensities and autocorrelation. It was tested for ice–water classification in the Fram Strait, showing that the C-band reliably reproduced the contours of ice edges, while the L-band had advantages in areas with thin ice/calm water. Suman Singha et al. [58] inputted the extracted feature vectors into a neural network classifier for pixel-wise supervised classification. The classification process highlighted matrix-invariant features like geometric strength, scattering diversity and surface scattering fraction as the most informative. The findings were consistent for both X-band and C-band frequencies, with minor variations observed for L-band. Furthermore, the authors of [59] explored the influence of seasonal changes and incidence angle on sea ice classification using an ANN classifier. The study concluded that in dry and cold winters, the classifier could adapt to moderate differences associated with the incidence angle. Additionally, it was found that the incidence angle dependency of backscatter remained consistent across various Arctic regions and ice types.

Juha Karvonen et al. [60] estimated ice concentration based on SAR image segmentation and MLP, combining high-resolution SAR images with lower-resolution radiometer data. In [61], they further demonstrated that MLP can estimate SIC from SAR alone, but the results were more reliable and accurate when SAR was combined with microwave radiometer data. Furthermore, they estimated the SIC and thickness in the Bohai Sea using dual-polarization SAR images from the 2012–2013 winter, AMRS 2 radiometer data and sea ice thickness data based on the High-resolution Ice Thickness and Surface Properties (HIGHTSI) model. Additionally, Yan et al. [62] demonstrated the feasibility of using the TDS-1 satellite data for neural network-based sea ice remote sensing using a satellite-based GNSS-R digital data acquisition system. It relied on a MLP neural network with back-propagation learning using an LM algorithm (800 inputs, 1 hidden layer with 3 neurons, and 1 output). In a recent study [63], it was shown that MLP outperformed LR in capturing the nonlinear decision boundaries, thus reducing misclassifications in certain cases. Additionally, MLP combined cognitive uncertainty prediction methods with arbitrary heteroscedastic uncertainty to allow estimation of uncertainty at each pixel location.

Overall, MLP has proven to be a valuable tool in sea ice remote sensing, providing accurate classification results and enabling the estimation of sea ice parameters. As research in this field continues, further advancements in MLP models and their integration with other data sources will contribute to a better understanding of sea ice dynamics, improved sea ice monitoring, and enhanced decision making for various applications related to sea ice.

### 2.2.4. Support Vector Machine (SVM)

Prior to the surge in popularity of deep learning, SVMs were the most favored model due to their solid mathematical foundation and the ability to achieve global optimum solutions (unlike linear models trained with gradient descent that may only converge to local optima). SVMs are commonly employed for binary classification tasks and are defined as linear classifiers that maximize the margin in the feature space.

The authors of [64] utilized backscattering coefficients, GLCM texture features and SIC as the basis for SVM-based sea ice classification. Experimental results demonstrated that SVMs exhibit stronger robustness against normalization effects compared to Maximum Likelihood (ML) results. Some cases [65–68] showcased the effectiveness of SVMs in distinguishing open-water areas from sea ice tasks. In [69], combining Kalman filtering,

GLCM and SVM yielded better sea ice accuracy compared to simple CNN models at that time. Yan et al. [70,71] proposed a simple yet effective feature selection (FS) approach and employed SVM classification, resulting in improved accuracy and robustness compared to NN, CNN and NN-FS approaches. Furthermore, experiments indicated that SVMs require less data storage and fewer tuning parameters.

Additionally, researchers have explored combining SVMs with other methods to enhance classification accuracy. For example, the authors of [72] integrated statistical distribution, region connection, multiple features and a SVM into the CRF model. Experimental comparisons revealed that the SVM-CRF achieved the best performance. Moreover, by utilizing Transductive Support Vector Machines (TSVMs) as the classifier had good performance on two hyperspectral images obtained from EO-1 [73].

In summary, SVMs were highly popular models in the field of sea ice classification before the rise of deep learning. They offer robustness, suitability for binary classification tasks and the potential for integration with other techniques, contributing to their effectiveness in accurately distinguishing sea ice from other classes. Furthermore, SVMs have advantages including lower data storage requirements and fewer tuning parameters.

### 2.2.5. Others

In addition to the commonly used machine learning methods mentioned above, decision tree (DT), LR, and k-means have also been used in ice classification tasks. The DT is commonly used to solve binary classification problems. For example, the authors of [74] employed a supervised classification model based on a DT to differentiate ice lakes from water ice using the radiometric and textural properties of Landsat 8 OLI multispectral data. Furthermore, Johannes Lohse et al. [75] utilized a DT for multi-class problems by decomposing them into a series of binary questions. Each branch of the tree separates one class from all other classes using a selected feature set specifically to that class. In the Fram Strait region, ice was accurately classified into categories such as grey ice, lead ice, deformed ice, level ice, grey-white ice and open water. Komarov et al. [76] modeled the probability of ice presence in the study area using LR. They automatically detected ice and open water from RADARSAT dual-polarized imagery. Additionally, based on the aforementioned modeling approach, they developed a multi-scale SAR ice–water inversion technique [77]. In [78], a multi-stage model was proposed for sea ice segmentation using superpixels. The preprocessing involved enhancing contrast and suppressing noise in high-resolution optical images. The segmentation results were refined through superpixel generation, K-means classification and post-processing.

Furthermore, various machine-learning algorithms have been combined to better extract sea ice. Wang et al. [79] proposed a two-round weight voting strategy in ensemble learning. In the first round of voting, six base classifiers, namely naive Bayes, DT, K-Nearest Neighbors (KNN), LR, ANN and SVM, were employed. Misclassified pixels were further refined through fine classification. Kim et al. [80] combined image segmentation, image correlation analysis and machine-learning techniques, specifically RF, extremely randomized trees and LR, to develop a fast ice classification model. Liu et al. [81] selected KNN and SVM classifiers for single-feature-based sea ice classification, while the classification of sea ice based on multiple feature combinations was performed using the selected KNN classifier. In [82], a Gaussian Markov Random Field model for automatic classification was introduced. The initial model parameters and the number of categories were determined by fitting the histogram of the imagery using a finite Gaussian mixture distribution. Experimental results show that it can achieve good classification effect.

### 2.2.6. Limitations

We aim to provide a more comprehensive discussion on the constraints associated with the application of machine-learning techniques in sea ice image segmentation. One notable limitation lies in the requirement for large labeled datasets for training. Many machine-learning algorithms, particularly deep-learning models, exhibit superior performance when

trained on large datasets. However, the collection and annotation of such datasets for sea ice imagery can be challenging and time-consuming. For instance, researchers in [83] highlighted the scarcity of labeled datasets as a major impediment to the development and evaluation of machine-learning algorithms for sea ice classification. Additionally, another limitation arises from the complexity and variability of sea ice characteristics, which can pose challenges for machine-learning models to generalize effectively across different regions and environmental conditions. This issue has been discussed extensively. In light of these limitations, further research efforts are needed to address dataset scarcity and enhance the robustness of machine-learning models for sea ice image analysis.

In summary, researchers have integrated different machine-learning algorithms to improve SIE. The two-round weight voting strategy and LR have demonstrated favorable classification performance. Combining image segmentation, correlation analysis and machine-learning techniques has facilitated the development of fast ice classification models. Additionally, the Gaussian Markov Random Field technique and self-supervised learning approaches have shown promise in SAR sea ice image classification. However, these approaches often involve manual feature extraction prior to network training, which can be a labor-intensive and time-consuming process. Additionally, when dealing with complex image scenes, the training process can become intricate and challenging.

## 2.3. Deep-Learning-Based Methods

Traditional approaches to sea ice classification rely heavily on manual feature extraction from remote-sensing images and the construction of classifiers. However, this methodology entails significant human and time costs, and often yields less accurate results in complex scenarios. In contrast, deep learning offers the ability to automatically learn and extract features, enabling more effective handling of sea ice classification tasks. Deep-learning methods, such as classification networks and semantic segmentation networks, have been widely applied in sea ice classification, showcasing remarkable performance in feature extraction and classification, thus significantly improving the accuracy of sea ice classification. In this section, we will discuss the applications of deep-learning methods in sea ice classification and explore the performance of different models in this domain, as shown in Figure 4.

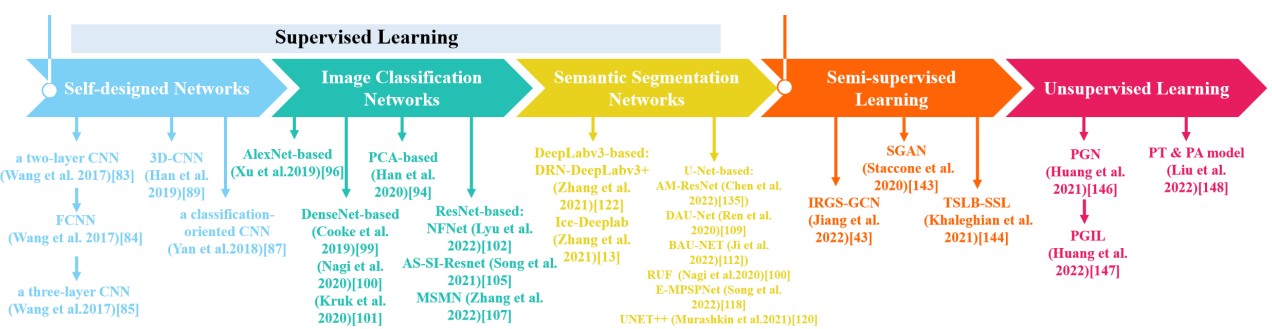

**Figure 4.** Chronological overview of the most relevant deep-learning-based SIE methods.

### 2.3.1. Supervised Learning

Early on, research generally used some simple CNN structures for sea ice classification. Wang et al. [84] were the first to employ CNN for SIC estimation from SAR images. Their work utilized a two-layer architecture consisting of convolutional and pooling layers, followed by a fully connected operation, eliminating the need for separate feature extraction or post-segmentation processing. The generated SIC maps exhibited an absolute average error of less than 10% compared to manually interpreted ice analysis charts. In [85], a fully convolutional neural network (FCNN) was proposed for estimating SIC from polarimetric SAR images. Experimental results showed slightly higher accuracy in SIC estimation using FCNN compared to CNN, along with additional computational efficiency.

In [86], a three-layer CNN with convolutional and pooling operations, as well as non-linear transformations, was constructed. This CNN demonstrated reduced differences and biases between ice concentration and labels compared to MLP or ASI algorithms, highlighting the superiority of CNNs. In [87], the CIFAR-10 CNN model was adapted to construct a CNN architecture, and experimental results demonstrated that CNN-based SIE achieved higher accuracy compared to traditional SVM methods. Yan et al. [88,89] designed a classification-oriented CNN for SIE and a regression-based CNN for SIC estimation. The CNN comprised $5\ 7 \times 7$ convolutional and pooling layers, followed by 2 fully connected layers. This was the first application of CNN technology to TDS-1 DDM data for SIE and SIC estimation. Compared to a NN, this approach exhibited improved overall accuracy and required fewer parameters and less data preprocessing. Han et al. [90] utilized GLCM to extract spectral and spatial joint features from hyperspectral sea ice images and constructed a 3D-CNN for sea-ice-type classification. In [91], CNN was employed for sea-ice-type classification based on Sentinel-1 SAR data, distinguishing between four categories: ice-free, young ice, FYI and old ice. Experimental comparisons with existing machine-learning algorithms based on texture features and RF demonstrated improved accuracy and efficiency. CNN-based SIC estimation was shown to outperform earlier estimation algorithms in [92]. Additionally, Malmgren-Hansen et al. [93] tested CNN under the scenario of disparate resolutions between Sentinel-1 SAR and AMSR 2 sensors and found that CNN was suitable for multi-sensor fusion with high robustness. Additionally, the integration of SE-Block into a 3D-CNN deep network was proposed in [94] to enhance the contribution of different spectra for sea ice classification. By optimizing the weights of various spectral features through the fusion of SE-Block, based on their respective contributions, the quality of samples was further improved. This approach enables superior accuracy classification of small-sample remote sensing sea ice images.

Given the significant progress in deep learning, a wide range of mature classification and segmentation networks have been developed. Researchers have successfully applied these existing networks to achieve accurate SIE. By building upon these established networks, they have been able to effectively extract sea ice from various data sources and achieve accurate results. In [95], a hyperspectral sea ice image classification method based on principal component analysis (PCA) was proposed. A comparison was made among SVM, 1D-CNN, 2D-CNN and 3D-CNN, showing promising results in sea ice classification with fewer training samples and a shorter training time. Xu et al. [96] employed transfer learning to extract features from patches using AlexNet and applied a softmax classifier, achieving an overall classification accuracy of 92.36% on test data. They also improved SIC estimation by augmenting the training dataset with more independent samples of undersampled classes [97]. The impact of transfer learning, data augmentation and input size on deep-learning methods for binary classification of sea ice and open water, as well as multi-classification of different types of sea ice, was further investigated in [98]. Subsequently, DenseNet [99] was introduced and demonstrated excellent performance on the challenging ImageNet database. In [100], DenseNet was employed to extract SIC from SAR images, achieving errors of 5.24% and 7.87% on the training and testing sets, respectively. DenseNet161 was used in [101], where multi-scale techniques were employed for automatic detection of the MIZ in SAR images. Analysis of the DenseNet prediction results by Kruk et al. [102] revealed that neural networks faced greater challenges in distinguishing different types of ice samples compared to differentiating between water and ice samples. Lyu et al. [103] obtained SIE and classification results for the first time from real polarimetric SAR data using the Normalizer-Free ResNet (NFNet) [104]. The Sea Ice Residual Convolutional Network (AS-SI-Resnet) was proposed in [105], and experimental results demonstrated its superiority over MLP, AlexNet and traditional SVM methods. The authors further considered spatial characteristics and temporal variations of sea ice and introduced long short-term memory (LSTM) networks to improve the accuracy of sea ice classification [106].

Building upon the outstanding performance of CNN in SIE tasks, researchers have further explored its application in larger datasets and research areas. Kortum et al. [107]

combined convolutional neural networks with dense conditional random fields (DCRF) and incorporated additional spatio-temporal background data to enhance model robustness and achieve multi-seasonal ice classification. Zhang et al. [108] developed a deep-learning framework called Multiscale MobileNet (MSMN), and experimental tests demonstrated an average improvement of 4.86% and 1.84% in classification accuracy compared to the SCNN and ResNet18 models, respectively. Singh Tamber et al. [109] trained a CNN using the binary cross-entropy (BCE) loss function to predict the probability of ice, and for the first time, explored the concept of augmented labels to enhance information acquisition in sea ice data.

In various domains, deep learning has made remarkable advancements in semantic segmentation in recent years. In particular, the U-Net network has been widely applied in various semantic segmentation tasks and has shown good segmentation performance. Researchers have also explored the application of the U-Net architecture in SIE. Ren et al. [110] proposed a U-Net-based model for sea ice and open water SAR image classification. This model can classify sea ice at the pixel level. Subsequently, the authors introduced a dual-attention mechanism, forming a dual-attention U-Net model (DAU-Net), which improved the segmentation accuracy compared to the U-Net model [111,112]. Kang et al. [9] improved the decoding network and loss function, achieving excellent results in the 2021 High-Resolution Challenge. A modified U-Net was used for automatic extraction of Antarctic glacier and ice shelf fronts. Ji et al. [113] constructed the BAU-NET by adding a batch normalization layer and an adaptive moment estimation optimizer to the U-Net. In addition, An FCN inspired by the U-Net architecture was applied to SIC prediction [114]. Radhakrishnan et al. [115] proposed a novel training scheme using curriculum learning based on U-Net to make the model training more stable. Wang et al. [116] stacked U-Net models to generate aggregated sea ice classifiers. Stokholm et al. [117] studied the effect of increasing the number of layers and receptive field size in the U-Net model on extracting SIC from SAR data. RES-UNET-CRF (RUF) was proposed in [118], which leverages the advantages of residual blocks and Convolutional Conditional Random Fields (Conv-CRFs), as well as a dual-loss function. Experimental results show that the proposed RUF model is more effective compared to U-Net, DeepLabV 3, and FCN-8. Song et al. [119] proposed a network called E-MPSPNet, which combines multi-scale features with scale-wise attention. Compared to mainstream segmentation networks such as U-Net, PSPNet, DeepLabV 3 and HED-UNet, the proposed E-MPSPNet performs well and is relatively efficient. UNET++ was proposed in [120], and it performs well in medical image segmentation tasks. Murashkin et al. [121] applied UNET++ to the task of mapping Arctic sea ice in Sentinel-1 SAR scenes. Feng et al. [122] proposed a joint super-resolution (SR) method to enhance the spatial resolution of original AMSR2 images. They used a DeepLabv3+ network to estimate SIC, which demonstrated good robustness in different regions of the Arctic at different times. In addition, Zhang et al. [123] combined semantic segmentation frameworks with histogram modification strategy to depict the disintegration frontier of Greenland's glaciers. It was found that the combination of histogram normalization and DRN-DeepLabv3+ was the most suitable. A hierarchical deep-learning-based pipeline was designed [124], which significantly improved the classification performance in numerical analysis and visual evaluation compared to previous flat N-way classification methods.

In addition, Colin et al. [125] conducted segmentation research on ten marine meteorological processes using the fully supervised framework U-Net, demonstrating the superiority of supervised learning over weakly supervised learning in both qualitative and quantitative aspects. Hoffman et al. [126] employed U-Net with satellite thermal infrared window data for Sea Ice Lead detection. An improved U-Net was used for glacier ice segmentation [127]. It introduced a new self-learning boundary-aware loss, which improved the segmentation performance of glacier fragments covering ice. CNN has not only been well-applied in SIE tasks but also used for extracting river and lake ice to achieve continuous monitoring of glacial lake evolution on Earth [128,129]. This research will provide references based deep learning for SIE tasks.

With the popularity and cost reduction of UAV technology, and considering its high spatiotemporal resolution, it has been widely applied in ice monitoring. It could fill the gap in

satellite imagery data to some extent. Zhang et al. [13,16] proposed ICENET and ICENETv2 networks for fine-grained semantic segmentation of river ice from UAV images captured in the Yellow River. ICENET achieved good results in distinguishing open water, surface ice and background. In addition to UAV imagery, some research has utilized in situ digital sea ice images captured by airborne cameras. Compared to large-scale satellite images, information recorded by airborne cameras has lower spatial scales, providing more detailed information about the formation of surrounding sea ice at higher resolutions. Dowden et al. [130] constructed semantic segmentation datasets based on photographs taken by the Nathaniel B. Palmer icebreaker in the Ross Sea of Antarctica. SegNet and PSPNet architectures were used to establish detailed baseline experiments for the datasets. In [131], an automated SIE algorithm was integrated into a mobile device. In [132], considering the impact of raindrops on the segmentation results of captured images, raindrop removal techniques were developed to improve the classification performance. In [133], a semantic segmentation model based on a conditional generative adversarial network (cGAN) was proposed. This model has good robustness and makes the effect of raindrops on the segmentation results smaller. In addition, a fast online shipborne system was developed and validated in [14] for ice detection and estimation of their locations to provide "ground truth" information supporting satellite observations. Ice-Deeplab [134] was developed to segment airborne images into three classes: Ocean, Ice and Sky. Zhao et al. [135] improved the U-Net network by introducing Vgg-16 and ResNet-50 for encoding, constructing the new networks VU-Net and RU-Net, and achieved good results in testing with mid-high-latitude winter sea ice images captured by airborne cameras. Furthermore, a multi-label sea ice classification model embedded with SE modules was used for airborne images [136], showing significant improvement in accuracy compared to machine-learning methods such as RF and gradient boosting decision tree [137].

Deep-learning techniques have also found application in predicting SIC from daily observations of passive microwave sensors such as SMMR, SSM/I and SSMI/S [138,139]. Chen et al. [140] utilized passive microwave and reanalysis data to quantitatively predict SIC, thereby providing not only navigational assurance for human activities in the Arctic but also valuable insights for studying Arctic climate change. Additionally, Gao et al. [141,142] have made significant contributions by employing collaborative representation and a transferred multilevel fusion network (MLFN) to detect and track sea ice variations from SAR images, which holds crucial importance for ensuring maritime safety and facilitating the extraction of natural resources.

### 2.3.2. Semi-Supervised Learning (SSL)

The current research on SIE is often limited by the scarcity of available datasets. To extract accurate information from large-scale datasets when only a limited number of labeled data are available, researchers have introduced SSL [143]. SSL is a technique that leverages unlabeled data to improve model performance. In the context of sea ice classification tasks. SSL can better utilize unlabeled sea ice images to enhance the model's classification accuracy. Staccone et al. [144] presented a SSL method based on generative adversarial networks (GANs) for sea ice classification. The approach leverages labeled and unlabeled data from two different sources to acquire knowledge and achieve more accurate results. Khaleghian [145] proposed a Teacher–Student label propagation method based on SSL (TSLP-SSL) to deal with a small number of labeled samples. Experimental results demonstrated its superior generalization capability compared to state-of-the-art fully supervised and three other semi-supervised methods, namely semi-GANs, MixMatch and LP-SSL. Jiang et al. [43] proposed a semi-supervised sea ice classification model (IRGS-GCN) that combines graph convolution to address this challenge. Furthermore, a weakly supervised CNN approach was proposed in [146] for ice floe extraction. This research leveraged a limited number of manually annotated ice masks as well as a larger dataset with weak annotations generated through a watershed segmentation model, requiring minimal effort. By effectively leveraging unlabeled or weakly labeled data, this method was able to build more accurate extraction models on limited labeled datasets.

### 2.3.3. Unsupervised Learning

Due to ongoing technological advancements, unsupervised learning has emerged as a promising approach for sea ice classification tasks. Taking advantage of the principle that SAR imagery can depict the electromagnetic properties of sea ice, Huang et al. employed a guided-learning approach based on physical characteristics, designing the structure and constraints of the models to better capture the scattering characteristics and information of sea ice in SAR imagery. By combining physical models, prior knowledge can be introduced into deep-learning models, enhancing their interpretability and generalization capability. In their work [147], the scattering mechanism was encoded as topic compositions for each SAR image, serving as physical attributes to guide CNNs in autonomously learning meaningful features. A novel objective function was designed to demonstrate the learning process of physical guidance. The unsupervised method achieved sea ice classification results comparable to supervised CNN learning methods. In another work [148], a novel physics-guided and injected learning (PGIL) unsupervised approach for SAR image classification was proposed. Compared to data-driven CNNs and other pre-training methods, PGIL significantly improved classification performance with limited labeled data. Furthermore, in [149], uncertainty was embedded into transfer learning to estimate feature uncertainty during the learning process. Experimental results demonstrated that this method achieved better sea ice classification performance.

This research all demonstrates that physics-guided learning can help address the issue of scarce sea ice data. Manual annotation of SAR imagery data is time consuming and expensive, making it challenging to obtain large-scale annotated data. However, physical characteristics can provide additional information to assist models in achieving more accurate classification and segmentation with limited labeled data. By leveraging physical models and prior knowledge, synthetic SAR imagery data can be generated for model training and optimization, thereby alleviating the problem of data scarcity. Therefore, future research can focus on achieving a more comprehensive and accurate understanding and classification of SAR imagery by combining physical characteristics with deep-learning methods.

### 2.3.4. Limitations

The application of deep learning in sea ice classification has certain limitations. One of these limitations is its dependence on labeled sea ice data for training, yet currently, there is a lack of large-scale and representative benchmark datasets. Additionally, the absence of large-scale models like SAM poses a challenge in determining whether it is feasible to conduct large-scale training across different regions and latitudes to adapt to varying SIC tasks. Furthermore, research on multi-source data fusion in SIC is relatively limited. The challenge lies in leveraging the complementary characteristics of different data sources to improve the accuracy of SIC. Multi-source data fusion can encompass remote-sensing images acquired from different sensors, meteorological data and oceanic observation data, among others. By integrating and analyzing these diverse datasets, more comprehensive and accurate sea ice information can be obtained.

## 3. Accessible Ice Datasets

According to the guidelines established by the World Meteorological Organization (WMO), sea ice can be classified in multiple ways, taking into account factors such as the stages of its growth process, its movement patterns, and the horizontal dimensions of its surface. The predominant classification method found in the literature is based on the developmental stages of sea ice, which encompass frazil ice, nilas ice, FYI and MYI. Additionally, some studies focus on specific tasks, such as the binary classification of open water and sea ice, as well as the multi-classification of different types of sea ice.

Currently, as researchers' interest in sea ice continues to grow, there is a rising availability of relevant datasets that are openly accessible. In order to meet the demands for further experimental evaluations and establish a standardized framework for future research, we have meticulously compiled a comprehensive database. This database encompasses all

currently available open-source SAR-based, optical-based, airborne camera-based and drone-based datasets. A total of 13 datasets have been collected, accompanied by detailed descriptions of their sources, as shown in Table 1. The emphasis is placed on key attributes such as sensor types, study areas, data sizes and partitioning methods, ensuring a comprehensive and structured resource for the research community.

*3.1. SAR-Based Datasets*

3.1.1. Radiation Characteristics of Sea Ice

SAR is the most commonly used active microwave data type and has been employed in 80% of SIC publications. The radar wavelength, polarization mode and incidence angle of SAR have significant impacts on the extraction performance. The specific parameters can be referred to in [7].

- **Radar wavelength** Much of the literature on sea ice classification has discussed the effectiveness of different radar wavelengths, including the Ku-band, X-band, L-band and C-band SAR. In summary, the X-band and Ku-band are suitable for winter sea ice monitoring, while the L-band offers advantages for summer sea ice monitoring. The C-band, which lies between the Ku-band and L-band, provides a balanced choice for sea ice monitoring across different seasons. Currently, many sea ice monitoring tasks opt for SAR in the C-band for research purposes. The authors of [150] demonstrate that, compared to the C-band, the L-band is more accurate in detecting newly formed ice.

- **Polarization mode** Polarimetric techniques offer valuable insights into sea ice identification by capturing more detailed surface information using polarimetric SAR. This leads to improved classification of different sea-ice-types. For instance, the distinctive rough or deformed surfaces of FYI result in higher backscattering coefficients in cross-polarization. Conversely, MYI, known for its stronger volume scattering, exhibits higher backscattering coefficients in both co-polarization and cross-polarization. Notably, Nilas ice, characterized by its smooth surface and high salinity content, demonstrates consistently low backscattering coefficients across both polarizations in radar observations.

- **Incidence angle** In many scattering experiments, the statistical characteristics of sea ice backscattering coefficients with respect to varying incidence angles can be observed distinctly. When a radar emits microwaves towards a calm open water surface, the echo signal becomes prominent when the incidence angle is close to vertical or extremely small. However, as the incidence angle increases, the backscattering from the sea surface weakens, resulting in a gradual reduction in surface roughness. Research has shown that at higher frequency bands, increasing the incidence angle improves the classification accuracy between sea ice and open water. Additionally, the backscattering coefficients during the melting period of sea ice are also influenced by the incidence angle. For instance, in HH-polarized data, the backscattering coefficients obtained at small incidence angles are significantly higher, and they exhibit a linear relationship with increasing incidence angles.

3.1.2. Datasets

- **SI-STSAR-7** [83] The dataset is a spatiotemporal collection of SAR imagery specifically designed for sea ice classification. It encompasses 80 Sentinel-1 A/B SAR scenes captured over two freeze-up periods in Hudson Bay, spanning from October 2019 to May 2020 and from October 2020 to April 2021. The dataset includes a diverse range of ice categories. The labels for the sea ice classes are derived from weekly regional ice charts provided by the Canadian Ice Service. Each data sample represents a 32 × 32 pixel patch of SAR imagery with dual-polarization (HH and HV) SAR data. These patches are derived from a sequence of six consecutive SAR scenes, providing a temporal dimension to the dataset.

- **The TenGeoP-SARwv dataset** [15] The dataset is built upon the acquisition of Sentinel-1A wave mode (WV) data in VV polarization. It comprises over 37,000 SAR image patches, which are categorized into 10 defined geophysical classes.
- **SAR WV Semantic Segmentation** The dataset is a subset of The TenGeoP-SARwv dataset. It consists of three parts: training, validation and testing. The images comprise 1200 samples and are stored as PNG format files with dimensions of $512 \times 512 \times 1$ uint8. The label data are stored as npy files, represented by arrays of size $64 \times 64 \times 10$, where each channel represents 1 of the 10 meteorological classes.
- **KoVMrMl** The dataset utilizes Sentinel-1 Interferometric Wide (IW) SAR data, including Single-Look Complex (SLC) and Ground Range Detected High-Resolution (GRDH) products in the HH channel. The GRDH images are annotated with 7 types of sea ice in patches of size $256 \times 256$. The H/$\alpha$ labeling is obtained by processing the dual-polarization SLC data using SNAP v9.0.0 software.
- **SAR-based Ice types/Ice edge dataset for deep learning analysis** The dataset is specifically compiled for sea ice analysis in the northern region of the Svalbard archipelago, utilizing annotated polygons as references. It encompasses a total of 31 scenes and contains 6 distinct classes. The dataset is organized into data records, referred to as patches, which are extracted from the interior of each polygon using a stride of 10 pixels. Each class is represented by patches of different sizes, including $10 \times 10$, $20 \times 20$, $32 \times 32$, $36 \times 36$ and $46 \times 46$ pixels.
- **AI4SeaIce [117]** The dataset consists of 461 Sentinel-1 SAR scenes matched with ice charts produced by the Danish Meteorological Institute during the period of 2018–2019. The ice charts provide information on SIC, development stage and ice form in the form of manually drawn polygons. The dataset also includes measurements from the AMSR2 microwave radiomete sensor to supplement the learning of SIC, although the resolution is much lower than the Sentinel-1 data. Building upon the AI4SeaIce dataset, Song et al. [119] constructed an ice–water semantic segmentation dataset.
- **Arctic sea ice cover product based on SAR [116]** The dataset is based on Sentinel-1 SAR and provides Arctic sea ice coverage data. Approximately 2500 SAR scenes per month are available for the Arctic region. Each S1 SAR image acquired in the Arctic has been processed to generate NetCDF sea ice coverage data. Each S1 image corresponds to an NC file. The spatial resolution of the SAR-derived sea ice cover is 400 m. The website has released the processing of S1 data obtained in the Arctic from 2019 to 2021 and has uploaded the corresponding sea ice coverage data.

*3.2. Optical-Based Datasets*

3.2.1. Common Optical Sensors

There are several types of optical sensors commonly used for ice classification:

- **MODIS** MODIS is an optical sensor widely used for ice classification. It is carried on the Terra and Aqua satellites. By observing the reflectance and emitted radiation of the Earth's surface, MODIS can provide valuable information about ice characteristics such as color, texture and spectral properties.
- **VIIRS** VIIRS is an optical sensor with multispectral observation capabilities, used for monitoring and classifying the Earth's surface. It provides high-resolution imagery and has applications in ice classification.
- **Landsat series** The Landsat satellites carry sensors that provide multispectral imagery for land cover classification and monitoring, including ice classification. Sensors such as OLI (Operational Land Imager) and TIRS (Thermal Infrared Sensor) on Landsat 8, as well as previous sensors like ETM+ (Enhanced Thematic Mapper Plus), have been extensively used in ice classification tasks.
- **Sentinel series** The European Space Agency's Sentinel satellite series includes a range of sensors for Earth observation, including multispectral and thermal infrared sensors. The multispectral sensor on Sentinel-2 is utilized for ice classification and monitoring,

while the sensors on Sentinel-3 provide information such as ice surface temperature and color.

- **HY-1 (Haiyang-1)** HY-1 also contribute to ice classification and monitoring. The HY-1 satellite is a Chinese satellite mission dedicated to oceanographic observations, including the monitoring of sea ice. The HY-1 satellite carries the SCA (Scanning Multichannel Microwave Radiometer) sensor, which operates in the microwave frequency range. This sensor can provide measurements of SIC, sea surface temperature and other related parameters. By detecting the microwave emissions from the Earth's surface, the SCA sensor can differentiate between open water and ice.

- **The VIIRS-based river ice maps** [151] The dataset furnishes daily updates on river ice conditions across continental scales, encompassing the northern basins of the United States and the entirety of Canadian territory. Segmentation of VIIRS imagery holds promise for facilitating the detection and mapping of river ice, while also enabling the generation of additional classes such as snow, water and clouds.

These optical sensors capture spectral information or radiation characteristics in different bands, enabling the acquisition of valuable data on ice morphology, types and distribution. They play a crucial role in ice classification and monitoring. These sensors are widely employed in remote sensing and Earth observation, providing valuable data for ice monitoring and research purposes.

### 3.2.2. Datasets

Compared to SAR-based datasets, there are fewer datasets based on optical imagery. To the best of our knowledge, there are currently two open-source optical imagery datasets available:

- **2021Gaofen Challenge** The dataset is based on HY-1 visible light images with a resolution of 50 m. The scenes cover the surrounding region of the Bohai Sea in China. The provided images have varying sizes ranging from 512 to 2048 pixels and consist of over 2500 images. Each image has been manually annotated at the pixel level for sea ice, resulting in two classes: sea ice and background. The remote-sensing images are stored in TIFF format and contain the R-G-B channels, while the annotation files are in PNG format with a single channel. In the annotation files, sea ice pixels are assigned a value of 255, and background pixels have a value of 0.

- **Arctic Sea Ice Image Masking** The dataset consists of 3392 satellite images of the Hudson Bay sea ice in the Canadian Arctic region, captured between 1 January 2016 and 31 July 2018. The images are acquired from the Sentinel-2 satellite and composed of bands 3, 4 and 8 (false color). Each image is accompanied by a corresponding mask that indicates the SIC across the entire image.

### 3.3. Datasets Based on Alternative Acquisition Methods

Ice classification datasets based on alternative acquisition methods include imagery captured by icebreakers and drones.

- **Airborne camera-based datasets** The dataset is constructed from GoPro images captured during a two-month expedition conducted by the Nathaniel B. Palmer icebreaker in the Ross Sea, Antarctica [130]. The video clips captured can be found at https://youtu.be/BNZu1uxNvlo, accessed on 1 January 2024. These images were manually annotated using the open-source annotation tool PixelAnnotationTool into four categories: ice, ship, ocean and sky. The dataset was divided into three sets, namely training, validation and testing, in an 8:1:1 ratio. Data augmentation was performed by horizontally flipping the images, resulting in a training dataset of 382 images.

- **River ice segmentation** [152] The dataset collects digital images and videos captured by drones during the winter seasons of 2016–2017 from two rivers in Alberta province: the North Saskatchewan River and the Peace River. The images in the dataset are segmented into three categories: ice, anchor ice and water. The training set consists

of 50 pairs, while the validation set includes 104 images; however, there are no labels available for the validation set.

- **NWPU_YRCC2 dataset** A total of 305 representative images were selected from videos and images captured by drones during aerial surveys of the Yellow River's Ningxia-Inner Mongolia section. These images contain 4 target classes and were cropped to a size of 1600 × 640 pixels. The majority of these images were collected during the freezing period. Each pixel of the images was labeled into one of four categories: coastal ice, drifting ice, water and other using Adobe Photoshop 2020 software. The dataset was split into training, validation and testing sets in a ratio of 6:2:2, comprising 183, 61 and 61 images, respectively.

These datasets provide valuable resources for training and evaluating ice classification algorithms using imagery from icebreakers and drones. They contribute to the development of accurate and robust models for ice classification, utilizing alternative data sources.

**Table 1.** The overview of the detailed description of the 12 datasets we collected.

| Type | Dataset | Data Source | Research Area | Task | Ref. | Download Link (accessed on 1 January 2024) |
|---|---|---|---|---|---|---|
| SAR-based | SI-STSAR-7 | Sentinel-1 A/B dual-polarization (HH and HV) in EW scan mode | cover the entire open ocean | Classified by: OW, NI, GI, GWI, ThinFI, MedFI and ThickFI | [83] | http://ieee-dataport.org/open-access/si-stsar-7 |
| | The TenGeoP-SARwv dataset | the WV in VV polarization from Sentinel-1A | over the open ocean | Classified by: Atmospheric Fronts, Biological Slicks, Icebergs, Low Wind Area, Micro Convective Cells, Oceanic Fronts, Pure Ocean Waves, Rain Cells, Sea Ice, Wind Streaks | [15] | https://www.seanoe.org/data/00456/56796/ |
| | SAR_WV Semantic Segmentation | Same as above | Same as above | Same as above | [125] | https://www.kaggle.com/datasets/rignak/sar-wv-semanticsegmentation |
| | KoVMrMl | Sentinel-1 IW SAR data, including SLC and GRDH products with HH channel | Belgica Bank, an ice-covered area along the north-east coast of Greenland | Classified by: Water, Young ice, FYI, Old ice, Mountains, Iceberg, Glaciers and Floating Ice | [147] | https://drive.google.com/file/d/1VK2geghwl_JUuEETntG_3_5rDBH8qnHN/view?usp=sharing |
| | SAR based Ice types/Ice edge dataset for deep learning analysis | Sentinel-1A EW GRDM | north of Svalbard | Classified by: Open Water, Leads with Water, Brash/Pancake Ice, Thin Ice, Thick Ice-Flat and Thick Ice-Ridged | — | https://dataverse.no/dataset.xhtml?persistentId=doi:10.18710/QAYI4O |
| | AI4SeaIce | The Sentinel-1 dual-polarization HH and HV, along with the PMR measurements from the AMSR2 instrument on the JAXA GCOM-W satellite | the waters surrounding Greenland | Sea ice concentration, developmental stages, and forms of sea ice | [117] | https://data.dtu.dk/articles/dataset/AI4Arctic_ASIP_Sea_Ice_Dataset_-_version_2/13011134/2 |

**Table 1.** *Cont.*

| Type | Dataset | Data Source | Research Area | Task | Ref. | Download Link (accessed on 1 January 2024) |
|------|---------|-------------|---------------|------|------|---------------------------------------------|
| | Arctic sea ice cover product based on spaceborne SAR | Sentinel-1 dual-polarization HH/HV data in EW mode | the Arctic | Arctic sea ice coverage data | [116] | https://www.scidb.cn/en/detail?dataSetId=771301999089025024 |
| Optical-based | 2021Gaofen Challenge | HY-1 visible light imagery with a resolution of 50 m | near the Bering Strait, China | Segmentation into sea ice and background | [9] | https://www.gaofen-challenge.com/challenge/competition/2 |
| | Arctic Sea Ice Image Masking | The Sentinel-2 satellite, composed of bands 3, 4, and 8 (false-color) | Hudson Bay sea ice in the Canadian Arctic | Segmented into different SIC categories | | https://www.kaggle.com/datasets/alexandersylvester/arctic-sea-ice-image-masking |
| | The VIIRS-based river ice maps | The following VIIRS I-bands are used: I01, I02, I03, and I05 | all rivers and waterbodies from western Alaska to the east coast of the US and Canada | Segmented into water, land, vegetation, snow, river ice, cloud, and cloud shadow | [151] | https://web.stevens.edu/ismart/land_products/rivericemapping.html |
| Airborne camera-based | Sea Ice Detection Dataset and Sea Ice Classification Dataset | GoPro images captured by the Nathaniel B. Palmer icebreaker | Ross Sea, Antarctica | automated detection of sea ice (ice, ocean, vessel, and sky) and classifying sea-ice-types (ocean, vessel, sky, lens artifacts, FYI, new ice, grey ice, and MYI) | [130] | https://youtu.be/BNZu1uxNvlo |
| Drone-based | River ice segmentation | The Reconyx PC800 Hyperfire professional game camera, and the Blade Chroma drone equipped with the CGO3 4K camera at the Genesee dock | two Alberta rivers: North Saskatchewan River and Peace River | Segmented into ice, anchor ice, and water | [152] | https://ieee-dataport.org/open-access/alberta-river-ice-segmentation-dataset |
| | NWPU_YRCC2 dataset | a fixed wing UAV ASN216 with a Canon 5DS visible light camera and a DJI Inspire 1 | the Ningxia–Inner Mongolia reach of the Yellow River | Segmented into: coastal ice, pack ice, water, and other | [16] | https://github.com/nwpulab113/NWPUYRCC2 |

## 4. Applications

Given the progress in SIE and classification technologies, obtaining accurate spatial distribution and dynamic changes of sea ice has become increasingly vital. Through careful analysis and evaluation, a multitude of valuable geographic information products have been developed. These products play a pivotal role in various domains, including weather forecasting [153], maritime safety [154], resource development [141] and ecological conservation [155]. In this section, we will delve into the specific applications, as shown in Figure 5.

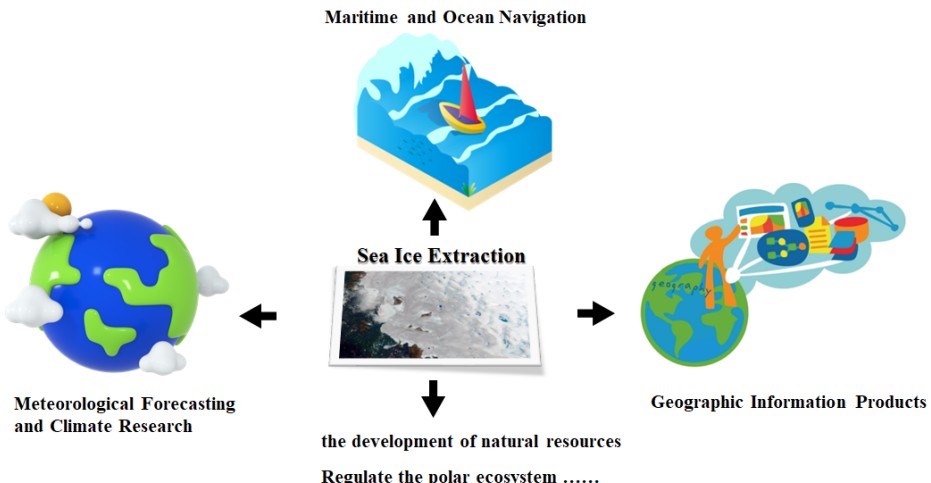

**Figure 5.** The extracted sea ice information finds significant applications in various domains, including meteorological forecasting and climate research, maritime navigation and geospatial information products.

### 4.1. Meteorological Forecasting and Climate Research

The results have significant applications in climate prediction. By utilizing remote-sensing techniques to extract and classify sea ice data, it becomes possible to improve the models that depict the interactions between the ocean and the atmosphere, further enhancing our understanding of sea ice response to climate change [156]. Analysis from research [153] reveals the potential value of sea ice observation data. The authors emphasize the regional variations in sea ice trends and highlight the lack of comprehensive records regarding marine connections. They utilize observation data to establish extensive Arctic and regional sea ice trends, enabling the identification and selection of climate models with optimal predictive capabilities on a global scale. These models subsequently provide more accurate predictions of future sea ice changes, which are closely linked to vital marine pathways in the Arctic region.

Furthermore, the extraction and classification of sea ice hold significant implications for monitoring climate change. This is due to the high albedo [157] of sea ice, which greatly alters the energy balance of the ocean. Additionally, sea ice exhibits low thermal conductivity, exerting a significant influence on the heat exchange between the ocean and the atmosphere. Thus, sea ice serves as a crucial indicator of climate change. Through regular extraction and classification of sea ice, we can monitor its temporal and spatial variations, analyze the trends of sea ice retreat and formation and provide data support for climate change research. Research outlined in [155] evaluates Arctic amplification and sea surface changes by observing the anomalies in Arctic sea ice extent, thickness, snow depth and ice concentration in comparison to the mean state during different periods (2011-2018).

Hence, the application of and classification is crucial for meteorological forecasting, climate prediction, and climate change monitoring. By utilizing remote-sensing techniques to extract and classify sea ice data, we can enhance the predictive capabilities of climate models, delve deeper into the interactions between sea ice and the climate system and assess and monitor the trends and impacts of climate change.

### 4.2. Maritime and Ocean Navigation

Accurate extraction and classification of sea ice data play a vital role in maritime and ocean navigation. By utilizing remote-sensing techniques to extract and classify sea ice information, it becomes possible to efficiently generate valuable products such as sea ice distribution maps, ice edge charts and route planning tools. These products serve as crucial aids for ships, enabling them to navigate safely and avoid ice-prone areas.

The Arctic Northeast Passage (NEP) has undergone remarkable changes in sea ice conditions, significantly impacting both the environment and navigational capabilities [158]. Research indicates a continued reduction in Arctic sea ice, leading to the shortening of trade routes in the Arctic Ocean and potentially affecting the global economy [159]. The authrs of [160], focusing on the Arctic NEP, have examined the influence of sea ice variations on the future accessibility of the route. While reduced sea ice has made it relatively easier for vessels to traverse the Arctic NEP, challenges and risks still persist. Another work [161] analyzed changes in sea ice volume and age, assessing the accessibility and navigable regions of the Arctic route.

Furthermore, the extent and thickness of sea ice hold significant importance for navigation, as emphasized in [162]. MYI, known for its thickness and hardness, poses substantial risks to ships. In contrast, younger and thinner ice enables icebreakers and regular cargo vessels to navigate more freely along ice-free coastal areas during the summer [163]. A recent study [164] investigated the impact of sea ice conditions. Similarly, research [165] revealed that sea ice thickness has a greater impact on vessel speed than ice concentration, underscoring its pivotal role in successful transit through the Arctic route. Therefore, future research endeavors should focus on enhancing the spatial and temporal resolution of sea ice monitoring to accurately evaluate the navigational capabilities of critical straits and regions.

Recent achievements have been made in this domain. A study [166] utilized high-quality, co-located satellite data and observation-calibrated reanalysis data to analyze sea ice changes along Arctic shipping routes. This research investigated the spatiotemporal distribution characteristics, melt/freeze timing and variations across trans-Arctic routes using datasets such as NSIDC SIC and daily Pan-Arctic Ice Ocean Modeling and Assimilation System (PIOMAS) SIT products. Additionally, by incorporating optimal interpolation sea surface temperature (SST) and SIC data, another study [167] examined the spatiotemporal distribution characteristics of SST and SIC above 60°N in the Arctic, along with their interrelationships. These findings hold crucial implications for Arctic shipping and sea ice forecasting, contributing to enhanced navigation and decision making in the region.

*4.3. Geographic Information Products*

In recent years, significant advancements have been made in utilizing remote-sensing techniques to generate geographic information products related to ice and polar regions. These applications encompass various aspects, including mapping, GIS and algorithmic approaches. The authors of [168] highlight the positive impact of Interferometric Synthetic Aperture Radar (InSAR) technology on Antarctic topographic mapping, not only at scales as small as 1:25,000 but also in thematic analysis and monitoring. By employing multiple radar images and D-InSAR techniques, it becomes possible to monitor subtle centimeter-level changes, offering tremendous potential for studying Antarctic glacier movement, mass balance and global environmental changes. In a similar vein, the authors of [3] demonstrate the production of polar remote-sensing products using very high-resolution satellite (VHRS) imagery, which proves to be an effective alternative to costlier aerial photographs or ground surveys. Moreover, Ref. [169] utilizes high-resolution ICESat laser altimetry to observe the dynamic changes in the grounding line of Greenland and Antarctic ice sheets, revealing a widespread thinning phenomenon across Greenland's latitudes and intensified thinning along critical Antarctic grounding lines. These findings hold crucial implications for Arctic shipping and sea ice forecasting, contributing to enhanced navigation and decision making in the region. Furthermore, Ref. [170] introduces the Ship Navigation Information Service System (SNISS), an advanced ship navigation information system based on geospatial data. SNISS offers a macroscopic perspective to develop optimal navigation routes for the Arctic NEP and provides ice image retrieval and automated data processing for key straits. Similarly, the authors of [171] developed RouteView, an interactive ship navigation system for Arctic navigation based on geospatial big data. By incorporating reinforcement learning and deep-learning technologies, RouteView calculates the optimal routes for the next 60 days and extracts sea ice distribution. These studies have the potential to enhance

the safety of vessels navigating the NEP and drive the development of augmented reality (AR) information-extraction methods. Arctic sea ice distribution maps serve as valuable aids for route planning, enabling vessels to avoid ice-covered areas and ensure sufficient water depth for safe passage. In addition, PolarView is a ship navigation and monitoring system specifically designed for polar regions. It offers real-time vessel positioning and navigation information, including sea ice coverage, ship route planning and hazard zone alerts. In the realm of path-planning optimization, a sophisticated maze path-planning algorithm with weighted regions has been proposed in research [154].

As remote-sensing techniques continue to advance and polar observation data become increasingly accessible, a variety of geographic information integration and visualization platforms have emerged. One notable platform is Quantarctica [172], which has been specifically designed as a comprehensive visualization platform for mapping Antarctica, the Southern Ocean and the islands surrounding Antarctica. It encompasses scientific data from nine disciplines, including sea ice, providing a wealth of information for researchers. Another significant resource is the International Bathymetric Chart of the Southern Ocean (IBCSO) [173], which offers detailed information about the bathymetry of the Southern Ocean. This dataset serves as a valuable resource for marine science research and the exploration of marine resources in the region. For terrain data in polar regions, ArcticDEM is a prominent system that enables terrain analysis, glacier research, hydrological modeling and more. Its comprehensive dataset contributes to a better understanding of the physical characteristics of the polar regions. To access a wide range of information about the polar regions, the ArcticWeb platform serves as a comprehensive polar information hub. It offers various resources including maps, satellite imagery, weather data and sea ice information. This integrated platform facilitates access to vital information for researchers, scientists and policymakers working in the polar regions. Additionally, there are online systems dedicated to sea ice monitoring and prediction. IceMap utilizes satellite data and numerical models to provide real-time sea ice coverage maps, thickness estimates and predictive simulations. It assists users in monitoring the state and trends of sea ice, providing valuable insights for various applications. For studying Arctic sea ice changes, the PIOMAS system offers simulation and analysis capabilities. It provides information on Arctic sea ice thickness, volume and distribution, which are crucial for climate research and analysis of ice conditions. In terms of monitoring snow and ice cover thickness in polar regions, the SNOWsat remote sensing system employs radar and laser altimetry data to deliver high-resolution measurements. These data are valuable for understanding snow depth and ice cover thickness, aiding in research related to climate change and polar ecosystems. Lastly, the Sea Ice Index, an online system provided by the U.S. National Snow and Ice Data Center, offers monitoring capabilities for global sea ice coverage and changes. It provides satellite-based sea ice indices and spatiotemporal distribution maps, enabling effective climate monitoring, environmental conservation and management of marine resources in polar regions. These systems collectively contribute to a comprehensive understanding of the polar regions and their dynamic characteristics. Moving forward, it is crucial to enhance the analytical capabilities of these systems by incorporating structured modeling of sea ice, enabling more sophisticated geographical analysis and providing better support for various applications in polar environments.

From glacier change observations to information system integration, and from ship navigation to route planning, these applications provide valuable data and tools for scientists, governments, policymakers and related industries, helping them better understand and manage sea ice resources. Additionally, scholars have conducted research on polar mapping and achieved significant results. Wang et al. [174] identified three commonly used map projection methods for the Antarctic region: Polar Stereographic Projection, Transverse Mercator Projection and Lambert Conformal Conic Projection, all of which are equal-angle projections. Figure 6 lists several commonly used projection visualizations of the Arctic region. The Quantarctica system utilizes the Antarctic Polar Stereographic projection EPSG:3031. Due

to the unique geographical position of polar regions, commonly used map projections have their limitations, and specific research is needed to address specific issues.

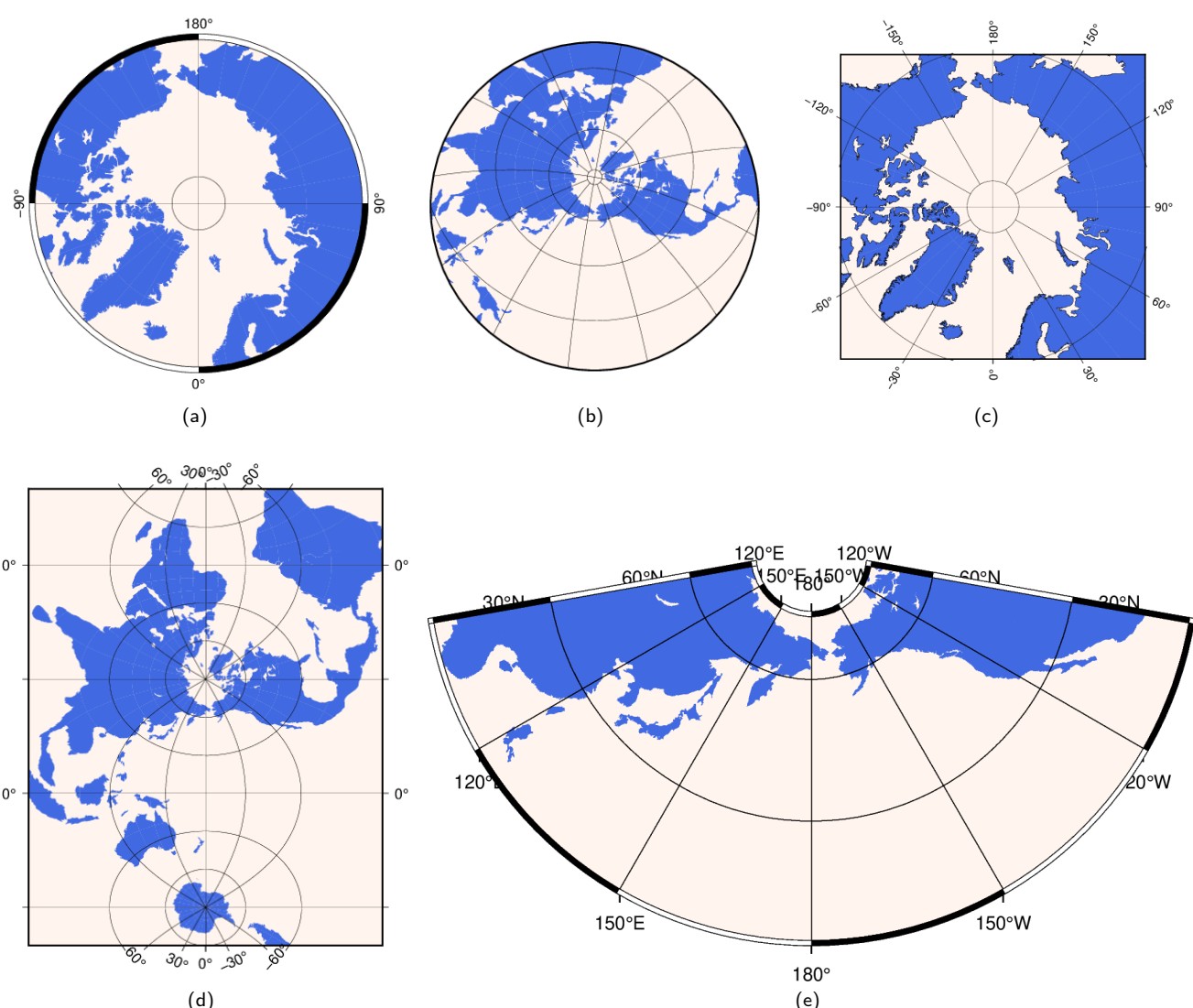

**Figure 6.** Several projection visualizations in the Arctic region: (**a**) The projection center is at the North Pole, characterized by a circular boundary. The map is symmetrically and uniformly distorted in all directions from the North Pole as the center. (**b**) The projection center is shifted away from the North Pole. The map still has a circular boundary, but the center is no longer the North Pole, and the distortion of the projection is not symmetric. (**c**) Rectangular maps are commonly used to display the entire polar region. (**d**) Vertical map. The Universal Transverse Mercator projection is used to simultaneously depict the North and South Poles. (**e**) The projection center is shifted, resulting in a non-global polar effect, with the coordinate range forming a sector-shaped area.

### 4.4. Others

Sea ice information is critical for the development of natural resources in coastal areas. Extracting and classifying sea ice can help assess its impact on activities such as fishing [175], oil and gas extraction [176] and submarine cable laying [177], providing important references for decision makers.

Sea ice is an essential component of the polar ecosystem. Its freezing and melting not only have a certain balancing effect on temperature changes in polar regions, but also affect the stability of ocean temperature, salinity and stratification, thereby impacting global ocean circulation [178]. Extracting and classifying sea ice can generate information such as

sea ice boundaries, ice–water interfaces and cracks, which are useful for ecological research and conservation efforts.

The results of and classification can be used in various fields of marine science [179,180], including ocean physics, marine biology and marine geology. By analyzing the characteristics and distribution of sea ice, changes and evolutionary processes of the marine environment can be inferred.

## 5. Challenges in Sea Ice Detection

There are several issues and challenges in SIE tasks. Firstly, a major problem is the limited availability of data sources, which restricts the accuracy and spatiotemporal resolution of SIC. The scarcity and discontinuity of existing data sources make it difficult to comprehensively capture and analyze sea ice features. Secondly, current SIC techniques have limited accuracy in complex sea ice conditions. Sea ice exhibits diverse variations in morphology, density, thickness and other characteristics, making it challenging for traditional algorithms to cope. Moreover, complex sea ice features such as cracks, ridges and leads undergo intricate changes, which are difficult to capture and represent using conventional methods. Additionally, there are limitations in the ability to detect underwater ice, making it challenging to obtain parameters such as its morphology and thickness. To address these issues, further exploration is needed in terms of detection methods, modeling approaches and mapping applications.

### 5.1. Exploration Methods Aspect

5.1.1. Multi-Sensor Integration

Current research primarily relies on optical imagery, SAR imagery or aerial photography captured by airborne cameras. Different sensors have their own characteristics and limitations in observing sea ice. A single sensor may not provide comprehensive information about sea ice. By introducing multi-sensor integration, the advantages of various sensors can be fully utilized to compensate for the limitations of a single sensor and obtain more comprehensive and accurate sea ice data. Multi-sensor integration can combine different technological approaches, such as microwave radar, optical sensors, acoustic techniques, etc., to acquire more comprehensive information about sea ice. For example, combining radar and optical sensor data enables simultaneous extraction of sea ice geometry and surface features, facilitating more precise and monitoring. Moreover, multi-sensor integration can also fuse data obtained from ground-based observations, satellite remote sensing, UAVs, and other platforms, providing multi-scale and multi-angle sea ice observations, thereby gaining a more comprehensive understanding of the spatiotemporal variations in sea ice.

In the Arctic Ocean, particularly in the Eastern Arctic, overcast sky conditions are prevalent, posing significant challenges for using optical satellite imagery to monitor sea ice. However, SAR imagery offers advantages of all-weather, all-day capability, unaffected by weather conditions, enabling the collection of clear, unobstructed images under any weather conditions. SAR imagery complements optical imagery by providing distinct texture features. Therefore, in the future, the integration of SAR and optical multimodal fusion methods can facilitate more comprehensive and accurate monitoring and analysis of sea ice.

Furthermore, establishing a continuous monitoring system using multiple sensors allows for dynamic monitoring and analysis of sea ice through long time series of remote sensing observations. By utilizing satellite remote sensing and other data sources, long-term monitoring of sea ice changes can be achieved to reveal seasonal and interannual variations. This enhances the reliability and consistency of data, enables multi-scale and all-weather sea ice observations and improves the capability of sea ice monitoring and prediction. These advancements provide more comprehensive and accurate data support for sea ice research and related applications.

5.1.2. Underwater Ice Detection

Currently, remote-sensing techniques are primarily used for employing remote-sensing sensors such as satellites, aircraft and UAVs to obtain image data of sea ice. Common remote-sensing techniques include optical remote sensing, SAR and multispectral remote sensing, which provide information on the spatial distribution, morphological features, cracks, and ice floes of sea ice. In addition, close-range images of sea ice can be acquired by mounting imaging devices on ships. Shipborne observations provide higher accuracy and local-scale sea ice information. Furthermore, UAVs equipped with sensors such as cameras and thermal infrared cameras enable high-resolution observations and measurements of sea ice. UAV technology offers high maneuverability and flexibility, allowing for more detailed information about sea ice [181,182].

However, remote-sensing methods are primarily suitable for surface detection and observation of sea ice, while direct remote-sensing detection of underwater ice, such as subsea ice caps, is relatively challenging. Due to the absorption and scattering properties of water, remote-sensing techniques are limited in their penetration and detection capabilities underwater. However, the detection of underwater ice is crucial for navigation and hydrographic surveying, as it can have significant implications for ship and navigation safety. The presence of underwater ice can lead to collisions, obstruction of navigation or structural damage to vessels. Therefore, accurate detection and localization of underwater ice are essential for safe navigation planning and guidance.

Some remote-sensing techniques and sensors can still provide some information about underwater ice under specific conditions. Sonar remote sensing is a technique that uses sound waves for detection and imaging in underwater environments. It can provide relevant information about underwater ice, such as the morphology of the ice bottom surface and ice thickness, by measuring the time and intensity of sound waves propagating in water. Sonar remote sensing finds widespread applications in the study of subsea ice caps and marine surveying. Additionally, technologies such as lasers and radars can also be used to some extent for underwater ice detection. Laser depth sounders can measure the distance and shape of underwater objects, providing information about ice thickness. Radar systems can penetrate to a certain depth underwater and detect the presence of underwater ice layers when operating at appropriate frequency bands.

*5.2. Model Approaches Aspect*

5.2.1. Multi-Source Data Fusion Model

The monitoring of sea ice primarily relies on SAR remote-sensing technology, which can penetrate meteorological conditions such as clouds, snowfall, and polar night to obtain high-resolution sea ice information. SAR also has the advantage of being sensitive to the structure and morphological changes of sea ice, enabling the identification and differentiation of different types of sea ice and providing more accurate monitoring and prediction of sea ice. There is also some research that utilizes optical remote-sensing technologies, such as visible light and infrared satellite imagery. However, optical remote sensing is limited under conditions of cloud cover, polar night and other factors, making it difficult to obtain clear sea ice information. Furthermore, due to the complexity and variability of sea ice, the limitations of a single optical remote-sensing technology can lead to misclassification and omission errors.

Therefore, some studies have fully considered the complementarity of optical and SAR data in sea ice classification and have fused the two to extract sea ice information in the study area. Li et al. [10] analyzed the imaging characteristics of sea ice in detail and achieved fusion by solving the Poisson equation based on Sentinel-1 and Sentinel-2 images to derive the optimal pixel values. Compared to the original optical images, the fused images exhibit richer spatial details, clearer textures and more diverse material textures and colors. The constructed OceanTDL 5 model is then employed for SIE.

In addition to directly fusing heterogeneous images, Han et al. [11] proposed a fusion of the features extracted from both sources. They first utilized an improved Spatial Pyramid

Pooling (SPP) network to extract different-scale sea ice texture information from SAR images based on depth. The Path Aggregation Network (PANet) was employed to extract multi-level features, including spatial and spectral information, of different types of sea ice from the optical images. Finally, these extracted low-level features were fused to achieve sea ice classification. In their work [12], they further introduced a Gate Fusion Network (GFN) to adaptively adjust the feature contributions from the two heterogeneous data sources, thereby improving the overall classification accuracy.

Han's work primarily focuses on feature-level fusion of SAR and optical images. In addition, input-level fusion and decision-level fusion have been demonstrated as effective methods [183–185], yielding favorable results in land use classification tasks. However, in the context of sea ice classification, it is crucial to consider the influence of different spectral bands on the radiation properties of sea ice. For instance, a simple approach involves replacing one of the R, G or B channels in the RGB image with a single SAR band. Through experimentation, it was found that replacing the B band yielded superior results, as the B band exhibits weaker texture characteristics while SAR better reflects the radiation properties of sea ice. Furthermore, another approach involves concatenating a single SAR band with the RGB three-channel image to form a four-channel image. However, during the model's pretraining process, there may be difficulties in loading certain weights, resulting in suboptimal outcomes.

### 5.2.2. Unsupervised Deep Learning

However, deep-learning methods currently face challenges in the classification of remote-sensing images, and one major challenge is the extensive manual annotation required. Additionally, accurate labeling of sea ice categories relies on expert knowledge, resulting in a scarcity of large-scale sea ice datasets for research purposes. The emergence of unsupervised deep learning presents a promising solution to this problem. By leveraging pre-training techniques such as transfer learning and self-supervised learning, unsupervised approaches can learn informative features for different sea-ice-types, enabling effective sea ice classification tasks.

Research generally focuses on specific regions of interest, such as the Greenland area. However, imagery exhibits variations across different regions, and sea ice distribution patterns differ as well. Consequently, testing the same model in different regions yields substantial discrepancies in the results. To tackle this challenge, the authors of [68] proposed the integration of texture features derived from GLCM into the extraction and classification of training samples. Unsupervised generation of training samples replaced the costly and labor-intensive process of manual annotation. Moreover, the method produced adaptable training samples that better accommodate the pronounced fluctuations in sea ice conditions within the Arctic MIZ. This concept has undergone initial testing using a subset of Gaofen-3 images. In response to the scarcity of labeled pixels in remote-sensing images, the authors of [186] present an effective approach for sea ice classification from two perspectives. Firstly, a feature extraction method is developed that extracts contextual features from the classification map. Secondly, an iterative learning paradigm is established. Experimental results demonstrate that with limited training data available, the training and classification of sea ice image representations with comprehensive exemplar representation under mutual guidance provide insights into addressing the scarcity of labeled sea ice data.

Therefore, in response to the limitations of annotated datasets in sea ice research, unsupervised deep learning emerges as a highly promising avenue. By directly extracting insights from unlabeled data itself, it serves as a powerful tool for automatic feature learning, representation learning and clustering. Unsupervised deep-learning methods exploit the intrinsic structures and patterns within sea ice imagery, enabling the automatic extraction of informative features without the reliance on external labels or manual feature engineering. Within the realm of sea ice classification tasks, unsupervised deep learning techniques, such as autoencoders, GANs and variational autoencoders (VAEs), excel at acquiring meaningful representations from unlabeled sea ice data. These approaches

discover similarities, textures, shapes and other discernible patterns inherent in sea ice images, thereby transforming them into valuable feature representations. Moreover, the utilization of extensive unlabeled sea ice data for training purposes expands the available dataset, consequently enhancing the generalizability and robustness of sea ice classification models across varying timeframes, locations and sensor conditions.

However, the application of unsupervised deep-learning methods to SIC tasks introduces certain challenges. Primarily, the absence of external labels as supervision signals may yield inaccurate or ambiguous feature representations. Therefore, it is imperative to design suitable objective functions and loss functions to guide the unsupervised learning process, ensuring the acquired features effectively facilitate the classification and analysis of sea ice images. Additionally, training unsupervised learning models may necessitate increased computational resources and time due to the involvement of complex network architectures and larger-scale datasets. Furthermore, evaluating the performance of unsupervised learning methods and conducting comparative analyses to discern the strengths and weaknesses of different approaches represent inherently challenging tasks in this domain.

### 5.2.3. Construct ICE-SAM Large Model

The Segment Anything Model (SAM) [187], originally designed for segmenting natural images, is capable of segmenting various objects. We applied this model to the task of sea ice classification, and the segmentation results are shown in Figure 7.

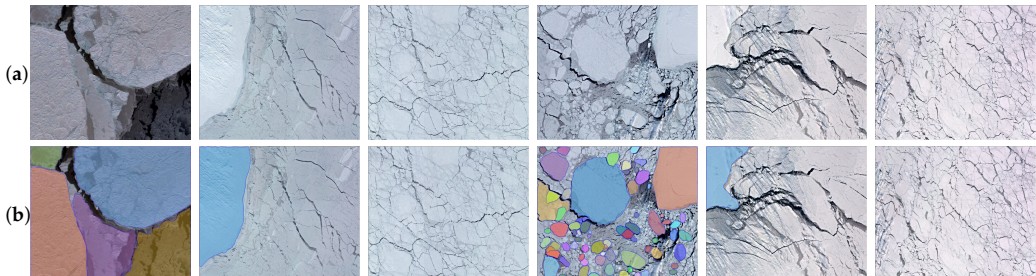

**Figure 7.** SAM segmentation results applied to Sentinel-2 imagery. (**a**) Sentinel-2 imagery and (**b**) SAM segmentation results. It can be observed that the first column accurately segments the image, the second and fifth columns can easily differentiate sea ice, the third and sixth columns do not perform segmentation and the segmentation result in the fourth column is excessively detailed.

SAM demonstrates high precision in the task of sea ice image segmentation, effectively distinguishing different types of sea ice. However, the model itself cannot directly determine the specific category names of the sea ice, i.e., it cannot associate the segmentation results with predefined sea ice categories. To address this issue, we try to introduce the CLIP model [188] as an auxiliary classifier, as it possesses the capability of joint understanding of images and text. We use the segmented sea ice image patches as inputs and compare them with a range of predefined sea ice category names. Through this comparative analysis, the CLIP model comprehends the connection between image content and category names, identifying the category that matches most. Consequently, we can accurately classify the sea ice image patches into their respective sea ice categories, obtaining specific category names for each sea ice region. Thus, the role of the CLIP model in sea ice image segmentation is to provide inference capability for sea ice category names. By leveraging its understanding of both images and text, the CLIP model establishes the association between segmentation results and category names, enabling us to acquire more comprehensive and detailed sea ice classification information. This approach allows for a more comprehensive understanding of sea ice features and attributes, providing more accurate data support for sea ice monitoring and research.

*5.3. Cartographic Applications Aspect*

5.3.1. Polar Geographic Information Systems (GIS)

Researchers have developed various GIS and tools specifically tailored for polar regions to support the processing, analysis, and visualization of polar environments and related data. In the early stages, a web-based GIS system [189] was developed, providing online access, exploration, visualization and analysis of archived sea ice data. Subsequently, systems such as PolarView, SNISS [170] and RouteView [171] were designed for polar navigation planning and ship navigation. These systems offer functionalities such as voyage planning, vessel position monitoring and channel information retrieval, utilizing real-time data and model analysis to facilitate safe and efficient navigation in polar waters. However, these systems have limited integration of information, and the analysis paths considered are relatively narrow, resulting in somewhat idealized outcomes that have only limited reference value. Furthermore, with the increasing availability of polar observation data, several geographic information integration and visualization platforms have emerged. For example, Quantarctica [172], (IBCSO) [173], ArcticDEM and ArcticWeb provide functionalities for visualizing polar geographic data, scientific data querying, map generation and analysis. Online systems dedicated to sea ice monitoring and prediction, such as IceMap, PIOMAS, SNOWsat and Sea Ice Index, offer real-time sea ice coverage data, thickness estimation and predictive simulations.

The aforementioned systems primarily encompass ship navigation and monitoring, sea ice monitoring and prediction, polar mapping and geospatial information display, ice thickness measurement, climate research and environmental protection. These GISs generally employ a layered architectural framework consisting of a data layer, an application layer and a user interface layer. The data layer is responsible for storing and managing various polar-related geographic data, generally organized and stored in databases or file systems. These data can originate from multiple sources such as satellite observations, remote-sensing imagery, marine surveys, meteorological stations and vessels. The application layer is dedicated to processing and analyzing polar geospatial data, providing various functionalities and services. Within these polar systems, the application layer includes functions such as sea ice monitoring and prediction, navigation planning and guidance, map creation and visualization and geospatial analysis and modeling. The functionalities within the application layer are typically implemented through algorithms, models and tools, enabling data processing, analysis and generating corresponding results and products. The user interface layer is responsible for presenting and displaying geospatial data, functionalities and results to users, facilitating interaction and visualization of the system's capabilities.

However, most existing systems primarily focus on data integration and visualization, lacking comprehensive geospatial analysis capabilities. In order to achieve geospatial analysis functions for polar regions (taking sea ice as an example), the architectural design and expansion of polar systems can be further improved. Here are some suggested feature enhancements and architectural directions:

- **Data Integration and Management.** Polar systems should integrate sea ice data from multiple sources and manage them in a unified and standardized manner. This includes satellite observations, marine measurements and more. To enable structured modeling and geospatial analysis, the data integration and management module should incorporate functionalities such as data cleansing, format conversion, quality control and metadata management.
- **Structured Modeling.** The system needs to develop algorithms and models for structured modeling of sea ice, transforming raw sea ice data into structured representations with geospatial information. This involves modeling sea ice morphology, density, thickness, distribution and the relationships between sea ice and other geographical features. The sea ice structured modeling module should consider the spatiotemporal characteristics of sea ice and establish associations with the geographic coordinate system.
- **Geospatial Analysis Capabilities.** The system should provide a wide range of geospatial analysis functions to extract useful geospatial information from the sea ice struc-

tured model. This may include spatiotemporal analysis of sea ice changes, thermodynamic property analysis, analysis of sea ice interactions with the marine environment and more. The geospatial analysis module should support various analysis methods and algorithms, along with interactive visualization and result presentation.

- **Real-time Data and Updates.** To ensure timeliness, the system should support real-time acquisition and updates of sea ice data. This can be achieved through real-time connections with data sources such as satellite observations, buoys, UAVs and more. Additionally, the system should possess efficient and scalable data storage and processing capabilities to handle large-scale data-processing requirements.

Future systems can further expand their architectural framework by incorporating technologies such as distributed computing, cloud computing and artificial intelligence to enhance system performance and scalability. Furthermore, strengthening data sharing, standardization and interoperability can facilitate data integration and functional consolidation among different systems, enabling a higher level of integration and collaborative work. These extended functionalities will enhance the overall performance and practicality of polar systems, providing comprehensive support for scientific research, navigation safety and environmental protection, among other domains.

### 5.3.2. Polar Map Projections

The unique shape and geographical attributes of the Earth's surface in polar regions make mapping challenging, hence research on polar cartographic projections has always been an important topic.

Specifically, Bian et al. [190] introduced the concept of complex variable isometric latitude based on the Gauss projection complex variable function. They overcame the limitations of traditional Gauss projections and established a unified and comprehensive "integrated representation" of Gauss projection in polar regions. Building upon this foundation, through rigorous mathematical derivations, they provided theoretically rigorous direct and inverse expressions for Gauss projection that can be used to fully represent polar regions, as well as corresponding scale factors and meridian convergence formulas. This approach addresses the problem of the impracticality of traditional Gauss projection formulas in polar regions and is of significant importance in improving the mathematical system of Gauss projection. It can be applied to the entire polar region and has important reference value for compiling polar maps and polar navigation [191]. Furthermore, the authors of [192] demonstrate that the non-singular Gauss projection formula for polar regions meets the requirements of continuous projection within the polar region, providing a theoretical basis for the production of polar charts. Due to its conformal property, Gauss projection can better determine directional relationships and is of significant reference value for the production of topographic maps along the central meridian in polar regions, and can be combined with the current need for polar navigation charts for the Arctic route. Gauss projection has advantages over sundial projection when applied to polar regions. Currently, most globally released Antarctic sea ice distribution maps are presented in a spherical projection, which cannot be directly used for mainstream tiled map publication. The authors of [193] convert polar azimuthal stereographic projection sea ice charts to the mainstream web Mercator projection map, and utilizes appropriate image resampling methods to generate tiles and store them with numbered tiles according to different scale levels, ultimately achieving the publication and sharing of sea ice image maps.

In recent years, there has been a relative lack of research on the latest developments in polar cartographic projections. The current major challenges include severe distortion of commonly used projection methods in polar regions and the difficulty of finding a suitable balance between equal area and equal angle properties. Additionally, polar regions generally possess highly complex data, such as sea ice distribution and ice sheet changes. Therefore, another challenge in polar projection is how to effectively visualize and present the geographical information of polar regions. To more effectively visualize and present

geographic information of the polar regions to meet the needs of different users, there are several potential research prospects and directions for future development, including:

- **Novel polar projection methods.** Researchers can continue to explore and develop new polar projection methods to address the existing issues in current projection methods. This may involve introducing more complex mathematical models or adopting new technologies such as machine learning and artificial intelligence to achieve more accurate and geographically realistic polar projections.
- **Multiscale and multi-resolution polar projections.** Polar regions encompass a wide range of scales, from local glaciers to the entire polar region, requiring map projections at different scales. Therefore, researchers can focus on how to perform effective polar projections at various scales and resolutions to meet diverse application requirements and data accuracy needs.
- **Dynamic polar projections.** The geographical environment in polar regions undergoes frequent changes, such as the melting of sea ice and glacier movements. Researchers can investigate how to address this dynamism by developing dynamic polar projection methods that can adapt to changes in the geographical environment, as well as techniques for real-time updating and presentation of geographic information.
- **Multidimensional polar projections.** In addition to spatial dimensions, data in polar regions also involve multiple dimensions such as time, temperature, and thickness. Researchers can explore how to effectively process and present multidimensional data within polar projections, enhancing the understanding of polar region changes and features.

## 6. Conclusions

This review provides a summary and overview of the methods used for SIE in the past five years, including conventional image classification methods, machine learning-based methods, and deep-learning-based methods. In addition, we have compiled a list of currently available open-source datasets for ice classification and segmentation, and explored the application aspects of from multiple perspectives. Finally, we have identified potential research directions based on the challenges encountered in detection methods, model approaches and cartographic applications.

**Author Contributions:** W.H. conducted most part of experiments and experimental analysis. Q.S. prepared dataset in this work. He also proposed the ideas of this work and gave many insight advises about this manuscript. A.Y. participated in CNN network design and model pruning. He also checked the quality of the dataset generated in this paper. W.G. and S.J. conducted part of the experiments, especially the ones in ablation studies. They also proofread this paper. Q.X. is the corresponding author of this work and provided experiment equipment. B.W. and C.Q. provided funds and part of the datasets. All authors have read and agreed to the published version of the manuscript.

**Funding:** This research was supported by the National Natural Science Foundation of China under grant Nos. 42101458,42171456,42130112,41901285 and the Fund Project of ZhongYuan Scholar of Henan Province of China under grant number 202101510001.

**Data Availability Statement:** Some or all data and code generated or used in this study are available from the corresponding author upon request. The data are not publicly available due to privacy restrictions.

**Acknowledgments:** The authors are grateful to the editors and the anonymous referees for their valuable comments and suggestions.

**Conflicts of Interest:** The authors declare no conflicts of interest.

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
