# Peer review of "Sea Ice Extraction via Remote Sensing Imagery: Algorithms, Datasets, Applications and Challenges"

_remotesensing, doi:10.3390/rs16050842_

Round 1
Reviewer 1 Report
Comments and Suggestions for Authors
This manuscript provides a comprehensive overview of utilizing remote sensing technology for analyzing sea ice. It delves into the intricate algorithms employed to process remote sensed imagery, focusing on methodologies like machine learning and image processing for accurate sea ice detection and classification. The manuscript explores a variety of datasets used in this field, detailing their sources, resolutions, and their pivotal role in extracting valuable insights about sea ice dynamics. Moreover, it sheds light on the practical applications of this information, ranging from climate studies to navigation safety and wildlife habitat assessment. Yet, it doesn't shy away from addressing the hurdles and limitations faced in this domain, discussing challenges such as varying ice types and the need for enhanced algorithms. Overall, the article serves as a comprehensive guide for researchers and professionals interested in leveraging remote sensing to study sea ice, encompassing its methodologies, datasets, applications, and the existing challenges within this evolving field. However, there are still some content and expression issue, please consider the following suggestions for minor revisions in your manuscript.
1)In the ‘2.1. Classical image segmentation methods’ section, the limitations mainly focus on threshold selection, mentioning the sensitivity of the detection threshold to image brightness and noise. Do all four methods mentioned involve threshold selection, or is it only the ‘3) Thresholding Method’? If only the ‘3) Thresholding Method’ needs select threshold, further limitations of other methods should be added.
2)Page 4, lines 131: The full names of VV and HH are missing.
3)Page 6, lines 214: Yang et al. used RF for lake ice extraction, which is different from the focus of this paper on sea ice extraction. The physical characteristics of lake ice and sea ice differ, making this content seemingly inappropriate.
4)Page 13, lines 571: The statement "which encompass frazil ice, nilas ice, FYI, and MYI" suggests sea ice can be classified into more than the mentioned four types. Clarify all the types according to the stage of sea ice before emphasizing the primary focus on the four types described.
5)Page 13, lines 590: The ‘Ku-band’ is missing from the list that includes X-band, L-band, and C-band SAR.
6)Page 20, lines 867: The content about the snow depth detector ‘SnowSAT’ is expressed as ‘SNOWsat’ on its official website. Please check again.
7)Page 23, lines 960-962: Consider referencing literature at this point.
8)Page 25, lines 1070: Based on Figure 6, consider setting qualifiers for sentence ‘effectively distinguishing different types of sea ice.’
9)Page 26, lines 1129: "Satellite observation" and "remote sensing imagery" are not strictly parallel concepts.
10)TABLE 1: In the ‘Arctic Sea Ice Image Masking’ dataset, the expression ‘Segmented into different SIC categories’ is too general; it would be beneficial to provide more specific details.
Author Response
Dear reviewer:
We are pleased to resubmit the revised version of our paper entitled " Sea Ice Extraction via Remote Sensing Imagery: Algorithms, Datasets, Applications and Challenges" (remotesensing-2840856). On behalf of all the contributing authors, we would like to express our sincere appreciations of the editors and all reviewers. These comments are all valuable and very helpful for revising and improving our paper, as well as significant to guide our research. We carefully reviewed the reviewers' comments and drew up a point-by-point response letter for the reviewers.
The reviewers' comments are laid out below in italics. We studied the comments carefully and made corrections based on the reviewers' comments. Our responses are given in normal font and corresponding changes/additions to the manuscript are given in blue. We uploaded the response letter (Response_to_Reviewers.pdf), an updated manuscript with blue highlighted text indicating changes (Revision.pdf) and a clean version (Revision_clean.pdf). We hope the revised version is now suitable for publication and look forward to hearing from you in due course.
Sincerely,
Wenjun Huang

Reviewer 2 Report
Comments and Suggestions for Authors
This review, and the paper presented by the authors, provides a summary and overview of the methods used for SIE in the past five years, including classical image segmentation methods based on machine learning and methods based on deep learning. The authors compiled a list of available open-source datasets for ice classification and segmentation and explored application aspects from several perspectives.
In the study carried out within the work, the authors identified the potential
research directions based on the challenges encountered in the detection methods, the models approached and within the cartographic applications.
Major current challenges include the severe distortion of projection methods commonly used in the polar regions and the difficulty of finding an appropriate balance between equal area and equal angle properties. In addition, polar regions generally possess very complex data, such as sea ice distribution and ice sheet changes.
Therefore, another challenge in polar projection is how to effectively visualize and present the geographic information of the polar regions. In order to more effectively visualize and present geographic information of the polar regions to meet the needs of different users, there are several potential research perspectives and future development directions, including:
• New methods of polar projection. Researchers can continue to explore and develop new polar projection methods to address existing problems in current projection methods. This may involve introducing more complex mathematical models or adopting new technologies such as machine learning and artificial intelligence to achieve more accurate and geographically realistic polar projections.
• Multiscale and multi-resolution polar projections. Polar regions span a wide range of scales, from local glaciers to the entire polar region, requiring map projections at different scales.
• Dynamic polar projections. The geographic environment in the polar regions has frequent changes, such as the melting of sea ice and the movements of glaciers, these can be highlighted in real time through the various methods presented and studied.
Researchers can investigate how to approach this dynamism by developing the dynamic polar environment, but also projection methods that can adapt to changes in the geographic environment and techniques for updating and presenting geographic information in real time.
• Multidimensional polar projections. In addition to spatial dimensions, data in polar regions also involve multiple dimensions such as time, temperature, and thickness.
I propose that the work be accepted and published in the journal.
Comments on the Quality of English LanguageThis review, and the paper presented by the authors, provides a summary and overview of the methods used for SIE in the past five years, including classical image segmentation methods based on machine learning and methods based on deep learning. The authors compiled a list of available open-source datasets for ice classification and segmentation and explored application aspects from several perspectives.
In the study carried out within the work, the authors identified the potential
research directions based on the challenges encountered in the detection methods, the models approached and within the cartographic applications.
Major current challenges include the severe distortion of projection methods commonly used in the polar regions and the difficulty of finding an appropriate balance between equal area and equal angle properties. In addition, polar regions generally possess very complex data, such as sea ice distribution and ice sheet changes.
Therefore, another challenge in polar projection is how to effectively visualize and present the geographic information of the polar regions. In order to more effectively visualize and present geographic information of the polar regions to meet the needs of different users, there are several potential research perspectives and future development directions, including:
• New methods of polar projection. Researchers can continue to explore and develop new polar projection methods to address existing problems in current projection methods. This may involve introducing more complex mathematical models or adopting new technologies such as machine learning and artificial intelligence to achieve more accurate and geographically realistic polar projections.
• Multiscale and multi-resolution polar projections. Polar regions span a wide range of scales, from local glaciers to the entire polar region, requiring map projections at different scales.
• Dynamic polar projections. The geographic environment in the polar regions has frequent changes, such as the melting of sea ice and the movements of glaciers, these can be highlighted in real time through the various methods presented and studied.
Researchers can investigate how to approach this dynamism by developing the dynamic polar environment, but also projection methods that can adapt to changes in the geographic environment and techniques for updating and presenting geographic information in real time.
• Multidimensional polar projections. In addition to spatial dimensions, data in polar regions also involve multiple dimensions such as time, temperature, and thickness.
I propose that the work be accepted and published in the journal.
Author Response

(The authors gave the same response as above.)

Reviewer 3 Report
Comments and Suggestions for Authors
1. Please clarify the rationale for focusing the review on studies starting from 2016, especially since references to pre-2016 studies are included. This discrepancy requires explanation.
2. The abstract should summarize the findings of the literature review and offer insights into future research directions.
3. Suggest changing "literature" to "studies" on line 52.
4. Confirm whether the scope includes studies since 2016, as earlier references are noted (Figure 1).
5. Ensure each statement that references a study includes the reference number at the end for clarity and consistency. Use a similar style to the first statement in line 159.
6. Recommend substituting "classical image segmentation" with "conventional image classification methods" or "statistical methods" for precision.
7. Suggest renaming section 2.1.4 to "Other Statistical Approaches".
8. Section 2.1.5 should provide more detailed discussions on limitations, supported by examples from cited studies.
9. Adopt a uniform reference similar to the first statement in line 159 throughout the document.
10. Consider revising the title of section 2.2.5 to better reflect its content.
11. The analysis of limitations for machine learning methods appears insufficient. Identify these limitations with literature examples.
12. Propose restructuring the machine learning section to differentiate between supervised and unsupervised studies.
13. For Figure 4 I suggest its removal if it does not add value.
14. Strongly recommend including a table summarizing potential research directions for clarity and accessibility.
15. Assert that a YouTube video, as mentioned in reference 13, should not be considered a dataset due to its non-academic nature.
16. I highly recommend adding a section to describe the advances of Deep Learning methods that could be applied in Sea Ice image segmentations. I think this is a key message that readers will be looking for.
Comments on the Quality of English LanguageModerate editing of English language required
Author Response

(The authors gave the same response as above.)

Reviewer 4 Report
Comments and Suggestions for Authors
The paper by Yu et al provides a review of recent progress in sea ice detection and sea ice classification using remote sensing datasets (mainly satellite, but also aerial imagery) with special emphasis on the modern state of machine learning and deep learning algorythms. This review paper is timely and coprehansive, the authors thorough cover the main aspects of the considered issue. In particular, the reference list consists of 195 papers, the majority of them describes classical, recent, and modern approaches for sea ice detection and sea ice classification. In my openion, this paper shoud be published after minor revision.
My general comment refers to lack of direct comparison of quality/accuracy/ of the discussed methods/applications/case studies in specific papers. Any step towards unification of methods and quality assessment would improve the review paper. Also, I recommend to add the information that overcast sky conditions, which are very typical for the Arctic Ocean, especially the Eastern Arctic, significantly hinder usage of optical satellite imagery for monitoring sea ice. Among minor comments I could point out the lack of capital letter at the beginnig of line 780. Also I would like to highlight that Figure 4 looks rather strange and is more typical for presentation rather than scientific paper.
Author Response

(The authors gave the same response as above.)

Round 2
Reviewer 3 Report
Comments and Suggestions for Authors
- Regarding the first comment please include the explanation in the manuscript. Authors can provide simple statistics on the number of studies before 2016, and after 2016 to support the literature review approach. I think that this point is very critical.
- Comment 11: Please clearly indicate how this response was included in the updated version of the manuscript, which section, and which lines. Thanks.
- Comment 14: I was not able to read the second statement of the comment as it is given in a language different than English.
- Section 3.3: Another recent open-source dataset hosted in GEE App, and could be downloaded for River ice segmentation:
https://web.stevens.edu/ismart/land_products/rivericemapping.html
Reference: https://www.mdpi.com/2072-4292/15/20/4896
Moderate editing of English language required
Author Response
Dear reviewer:
We are pleased to resubmit the revised version of our paper entitled " Sea Ice Extraction via Remote Sensing Imagery: Algorithms, Datasets, Applications and Challenges" (remotesensing-2840856). On behalf of all the contributing authors, we would like to express our sincere appreciations of the editors and all reviewers. These comments are all valuable and very helpful for revising and improving our paper, as well as significant to guide our research. We carefully reviewed the reviewers' comments and drew up a point-by-point response letter for the reviewers.
The reviewers' comments are laid out below in italics. We studied the comments carefully and made corrections based on the reviewers' comments. Our responses are given in normal font and corresponding changes/additions to the manuscript are given in blue. We uploaded the response letter (Response_to_Reviewers.pdf) and an updated manuscript with blue highlighted text indicating changes (Revision.pdf). We hope the revised version is now suitable for publication and look forward to hearing from you in due course.
Sincerely,
Wenjun Huang
